# 7-Dehydrocholesterol-derived oxysterols cause neurogenic defects in Smith-Lemli-Opitz syndrome

**Hideaki Tomita, Kelly M Hines[†], Josi M Herron, Amy Li, David W Baggett[‡], Libin Xu***

Department of Medicinal Chemistry, University of Washington, Seattle, United States

**\*For correspondence:**
libinxu@uw.edu

**Present address:** [†]Department of Chemistry, University of Georgia, Athens, United States; [‡]Department of Structural Biology, St. Jude Children's Research Hospital, Memphis, United States

**Competing interest:** The authors declare that no competing interests exist.

**Abstract** Defective 3β-hydroxysterol-$\Delta^7$-reductase (DHCR7) in the developmental disorder, Smith-Lemli-Opitz syndrome (SLOS), results in a deficiency in cholesterol and accumulation of its precursor, 7-dehydrocholesterol (7-DHC). Here, we show that loss of *DHCR7* causes accumulation of 7-DHC-derived oxysterol metabolites, premature neurogenesis from murine or human cortical neural precursors, and depletion of the cortical precursor pool, both in vitro and in vivo. We found that a major oxysterol, 3β,5α-dihydroxycholest-7-en-6-one (DHCEO), mediates these effects by initiating crosstalk between glucocorticoid receptor (GR) and neurotrophin receptor kinase TrkB. Either loss of *DHCR7* or direct exposure to DHCEO causes hyperactivation of GR and TrkB and their downstream MEK-ERK-C/EBP signaling pathway in cortical neural precursors. Moreover, direct inhibition of GR activation with an antagonist or inhibition of DHCEO accumulation with antioxidants rescues the premature neurogenesis phenotype caused by the loss of *DHCR7*. These results suggest that GR could be a new therapeutic target against the neurological defects observed in SLOS.

## Editor's evaluation

This is an important paper that provides a conceptual framework for understanding how altered lipid metabolism can impact brain development. The authors use mouse models and human iPSCs to provide a convincing mechanistic explanation of how mutations in a key enzyme in cholesterol biosynthesis lead to a neurodevelopmental disorder.

## Introduction

Brain is rich in cholesterol, contributing to 25% of total cholesterol in the human body, and nearly all cholesterol in the brain is synthesized locally (*Dietschy and Turley, 2004*). Therefore, dysregulation of cholesterol metabolism in CNS can potentially cause significant defects in CNS development and functions. Indeed, enzymatic deficiencies in cholesterol biosynthesis cause a number of inherited diseases with severe neurodevelopmental phenotypes (*Porter and Herman, 2011*).

Smith-Lemli Opitz syndrome (SLOS) is an autosomal recessive, neurological and developmental disorder characterized by multiple developmental defects, such as distinctive facial features, cleft palate, microcephaly and holoprosencephaly, as well as severe intellectual impairment and behavioral problems (*Porter and Herman, 2011*; *Thurm et al., 2016*). Notably, SLOS patients display a high incidence (>50%) of autism spectrum disorders (*Bukelis et al., 2007*; *Sikora et al., 2006*; *Tierney et al., 2006*). SLOS is caused by mutations in the 3β-hydroxysterol-$\Delta^7$-reductase gene (*DHCR7*), which encodes the enzyme that converts 7-dehydrocholesterol (7-DHC) to cholesterol in the final step of the cholesterol biosynthesis pathway (*Fitzky et al., 1998*; *Tint et al., 1994*; *Wassif et al., 1998*). Defective DHCR7 resulting from the mutations leads to deficiency in cholesterol and accumulation of 7-DHC

in tissues and fluids of affected individuals (*Tint et al., 1994*; *Tint et al., 1995*). 7-DHC was found to be highly reactive toward free radical oxidation, leading to the formation of its oxidative metabolites, that is oxysterols (*Xu et al., 2009*; *Xu et al., 2010*; *Xu et al., 2013*; *Xu et al., 2011*). 7-DHC-derived oxysterols can exert cytotoxicity in neuronal cells, induce gene expression changes, and increase formation of dendritic arborization from cortical neurons (*Korade et al., 2010*; *Xu et al., 2012*). Interestingly, increased dendrite and axon formation has also been observed in neurons isolated from *Dhcr7*-KO (*Dhcr7*^-/-^) mouse brain (*Jiang et al., 2010*). These studies suggest that 7-DHC-derived oxysterols may be underlying the alterations in neuronal processes in SLOS.

Cholesterol, 7-DHC, and oxysterols derived from both have been found to play important roles in modulating signaling pathways in developing tissues and organs, such as Hedgehog (Hh) and Wnt signaling pathways (*Byrne et al., 2016*; *Corcoran and Scott, 2006*; *Huang et al., 2018*; *Myers et al., 2017*; *Porter et al., 1996*; *Raleigh et al., 2018*). Related to SLOS, ring-B oxysterols derived from 7-DHC oxidation inhibit Smo in the Hh pathway (*Sever et al., 2016*). On the other hand, cholesterol was recently found to selectively activate canonical Wnt signaling over non-canonical Wnt signaling (*Sheng et al., 2014*). However, in human induced pluripotent stem cells (hiPSCs) derived from SLOS patient fibroblasts, accumulation of 7-DHC was found to inhibit Wnt/β-catenin pathway, which contributes to the precocious neuronal specification in SLOS neural progenitors (*Francis et al., 2016*). Depending on the position of the oxidation, oxysterols have also been shown to bind and activate other signaling molecules and nuclear receptors, including estrogen receptors, liver X receptors, and glucocorticoid receptor (GR), and thus, play important roles in neurodevelopment and diseases (*DuSell et al., 2008*; *Theofilopoulos et al., 2013*; *Voisin et al., 2017*). However, it is unknown how alteration of sterol composition influences neural stem cell/progenitor behaviors during cortical development, and it remains elusive whether neural defects of SLOS are due to deficiency in cholesterol or accumulation of 7-DHC or its oxysterols.

Here, we examine the effects of *DHCR7* mutations in developing neural precursors, focusing on the cerebral cortex. We demonstrate that 7-DHC-derived oxysterols start to accumulate at embryonic day 12.5 (E12.5) and continue to increase at E14.5 and E16.5, and that loss of *DHCR7* causes decreased proliferation and self-renewal of mouse cortical neural precursors and aberrant premature neurogenesis in both mouse and human neural progenitor cells (NPCs). We then provide evidence that a 7-DHC-derived oxysterol, 3β,5α-dihydroxycholest-7-en-6-one (DHCEO), activates GR and the downstream receptor tyrosine kinases (RTKs)-mediated neurogenic signaling through TrkB, and in doing so, promotes premature NPC differentiation and perturbs neuronal positioning. Either inhibition of GR activation with an antagonist or inhibition of DHCEO accumulation with antioxidants rescues the premature neurogenesis defect.

## Results

### Dhcr7 is expressed in embryonic cortical precursors and neurons of developing murine cortex

To understand the role of *Dhcr7* in neural development, we studied murine developing cortical precursors during embryonic neurogenesis. We first analyzed the expression of *Dhcr7* mRNA during mouse cortical development. RT-PCR analysis showed expression of *Dhcr7* in embryonic day 11 (E11) through postnatal day 0 (P0) cortex (*Figure 1A*). Quantitative PCR (qPCR) also suggested the stable expression of *Dhcr7* from E11 to E18 (*Figure 1B*). Western blot analysis further confirmed the presence of Dhcr7 protein in the developing cortex (*Figure 1C*). Immunostaining showed that Dhcr7 was broadly expressed in the developing cortex and was detectable in precursors in the E12 ventricular and subventricular zones (VZ/SVZ) (*Figure 1D*). Immunostaining of cortical sections of *Dhcr7*^-/-^ mouse brain at E12 with this particular Dhcr7 antibody confirmed the antibody specificity against Dhcr7. To further establish the expression of Dhcr7 in neural lineage cells, we analyzed cortical precursor cultures from E12.5, which consists of proliferating radial glial cells that generate neurons in culture (*Figure 1E*). In the E12.5 cortical precursor cultures, Dhcr7 + cells were co-labeled with the precursor markers, Sox2 and Pax6. Dhcr7 was also co-labeled with the neuronal marker, βIII-tubulin, and the neural lineage marker, Nestin. The results suggested that Dhcr7 was consistently expressed in neural lineage cells, which is consistent with our in vivo analysis.

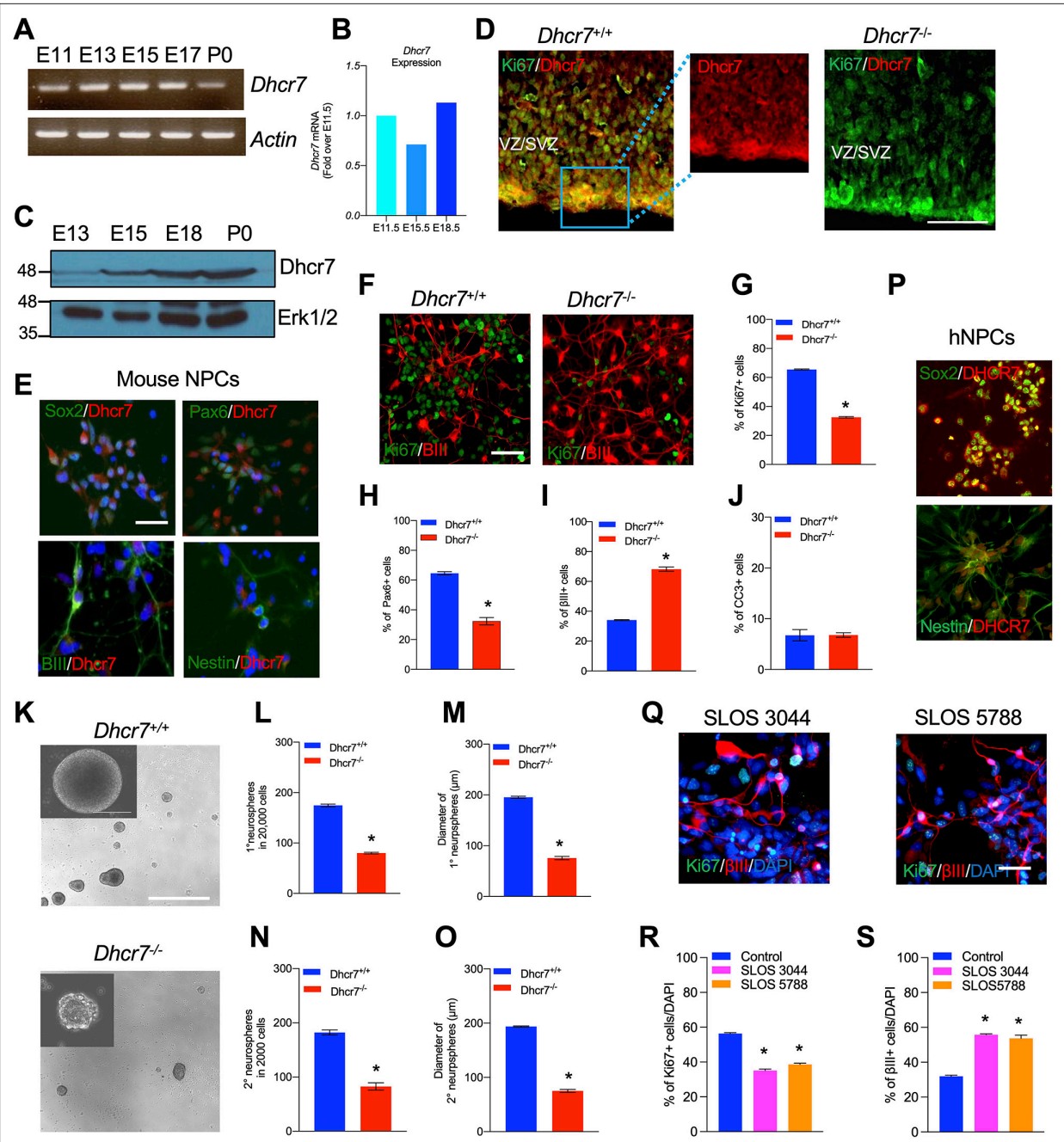

**Figure 1.** Loss of *Dhcr7* alleles causes decreased proliferation and increased neurogenesis in murine and human neural progenitor cells. (**A**) RT-PCR for *Dhcr7* mRNA in the E11.5 to P0 cortex. *β-actin* mRNA was used as the loading control. (**B**) qRT-PCR for *Dhcr7* mRNA in the E11.5 to E18.5 cortex. Data is expressed as fold change over E11.5 cortex. (**C**) Western blot of Dhcr7 in total cortical lysates from E13.5 to P0. The blot was re-probed for Erk1/2 as a loading control. (**D**) Images of E13.5 *Dhcr7^{+/+}* (*Dhcr7*-WT, left panel) and *Dhcr7^{-/-}* (*Dhcr7*-KO, right panel) cortical sections immunostained for Dhcr7 (red). The subventricular/ventricular zone (SVZ/VZ) is denoted. The right panels show the image of the boxed area. Scale bar = 100 µm. (**E**) Images of cultured mouse cortical precursors immunostained for Dhcr7 (red) and Sox2, Pax6, βIII-tubulin and Nestin (green) and counterstained with DAPI (blue). Scale bar = 50 µm. (**F–J**) E12.5 cortical precursors from single *Dhcr^{+/+}* and *Dhcr7^{-/-}* embryos were cultured for 3 days and analyzed. (**F**) Cells were immunostained for Ki67 (green) and βIII-tubulin (red) after 3 days and quantified for the proportions of Ki67 + (**G**), Pax6+ (**H**), βIII-tubulin+ (**I**) and CC3 + cells (**J**). Scale bar = 50 µm. *, p<0.001; n=3 embryos per genotype. (**K–O**) E13.5 cortical precursor cells from single *Dhcr7^{+/+}* or *Dhcr7^{-/-}* embryos were cultured as primary neurospheres (**K**) and the number and diameter of primary neurospheres were quantified 6 days later (**L, M**). Equal numbers of primary neurospheres were then passaged, and the number and diameter of secondary neurospheres were quantified 6 days later (**N, O**). Representative images of *Dhcr7^{+/+}* and *Dhcr7^{-/-}* neurospheres are shown as inserts in the left corner. *, p<0.001; n=3 embryos per genotype. Scale Bar = 100 µm. (**P–S**) Loss of *DHCR7* alleles causes decreased proliferation and increased neurogenesis in human cortical precursors. (**P**) Images of cultured human cortical

*Figure 1 continued on next page*

*Figure 1 continued*

precursors immunostained for Dhcr7 (red) and Sox2 or Nestin (green). (**Q–S**) Human SLOS patient-derived (SLOS 3044 and SLOS 5788) and unaffected individual (Control)-derived cortical precursors were cultured for 3 days and analyzed. (**Q**) Cells were immunostained for Ki67 (green) and βIII-tubulin (red) after 3 days and quantified the proportions of Ki67 + (**R**) and βIII-tubulin + cells (**S**). Scale Bar = 50 μm. *, p<0.001; n=3 biological replicates per genotype.

The online version of this article includes the following source data and figure supplement(s) for figure 1:

**Source data 1.** Related to *Figure 1A*.

**Source data 2.** Related to *Figure 1C*.

**Figure supplement 1.** Characterization of the pluripotency of SLOS-derived human iPSCs.

**Figure supplement 1—source data 1.** Related to *Figure 1—figure supplement 1B*.

**Figure supplement 1—source data 2.** Related to *Figure 1—figure supplement 1D*.

## Loss of Dhcr7 alleles causes decreased proliferation and increased neurogenesis in murine and human NPCs

SLOS is characterized by neurodevelopmental defects, such as microcephaly. To ask if *Dhcr7* plays important roles in neural precursor (progenitor) development as seen in human patients with SLOS, we intercrossed *Dhcr7*$^{+/-}$ mice and prepared single embryo cultures from E12.5 *Dhcr7*$^{-/-}$ (knockout or KO) or *Dhcr7*$^{+/+}$ embryos. Cultures were immunostained for Ki67 and βIII-tubulin 3 days after plating, which revealed that loss of *Dhcr7* caused a significant decrease in the proportion of Ki67 + and Pax6+precursors whereas increased the proportion of βIII-tubulin+neurons (*Figure 1F–I*). Immunostaining for cleaved caspase 3 (CC3) showed that the loss of *Dhcr7* alleles did not affect the survival of cortical precursor cells in culture (*Figure 1J*). These results suggested that loss of *Dhcr7* leads to premature neurogenesis and decreased proliferation of cortical precursors.

To determine if Dhcr7 is important for the proliferation and self-renewal of cortical precursors, we performed neurosphere assays, which assess if sphere-forming precursors can self-renew and generate new spheres. E13.5 cortical precursors from *Dhcr7*$^{-/-}$ and *Dhcr7*$^{+/+}$ embryos were cultured in the presence of FGF2 and EGF. The number and diameter of spheres were measured 7 days post-plating (*Figure 1K–O*). Significantly fewer number of and smaller neurospheres were generated from *Dhcr7*$^{-/-}$ embryonic cortices compared to its *Dhcr7*$^{+/+}$ littermates (*Figure 1L–M*). These results were consistent with the reduced proportion of Ki67 + precursors in the adherent cultures. Following the formation of the primary neurospheres, these spheres were triturated and re-plated at an equal density to form the secondary neurospheres. The results showed that there was an approximately twofold decrease in the number and diameter of secondary spheres from *Dhcr7*$^{-/-}$ cortical precursors relative to *Dhcr7*$^{+/+}$ (*Figure 1N–O*). Thus, the loss of *Dhcr7* alleles disrupts the proliferation and self-renewal of cortical precursors.

We then asked whether *DHCR7* was also necessary for the neurogenesis of human stem cell-derived NPCs. To examine the function of DHCR7, we generated human induced pluripotent stem cells (hiPSCs) from two lines of SLOS patient fibroblasts and one line of unaffected (Control) human fibroblasts and verified their pluripotency and stemness as described in Methods (*Figure 1—figure supplement 1*; *Okita et al., 2011*; *Yu et al., 2007*). The SLOS hiPSCs were then differentiated into hNPCs as described previously. Immunoreactivity of DHCR7 was detected in almost all Control hNPCs in these cultures with Sox2 + and Nestin + cells (*Figure 1P*). To examine neural differentiation, the SLOS and Control NPCs were differentiated in cultures and immunostained 4 days later for Ki67 and βIII-tubulin for proliferating precursors and newborn neurons, respectively (*Figure 1*). Notably, the proportion of Ki67 + precursors was significantly decreased whereas the proportion of βIII-tubulin+neurons were increased in SLOS NPC cultures, similar to the phenotype seen in murine cortical precursors from *Dhcr7*$^{-/-}$ embryos. Taken together, DHCR7 is involved in the proliferation and differentiation of human NPCs.

## Cholesterol precursor 7-DHC and 7-DHC-derived Oxysterols are accumulated in Dhcr7$^{-/-}$ developing cortex and human NPCs

7-DHC is highly susceptible to free radical oxidation (*Xu et al., 2009*), which leads to the formation of numerous oxysterols in cells and tissues (*Xu et al., 2013*; *Xu et al., 2011*; *Figure 2A*). Liquid

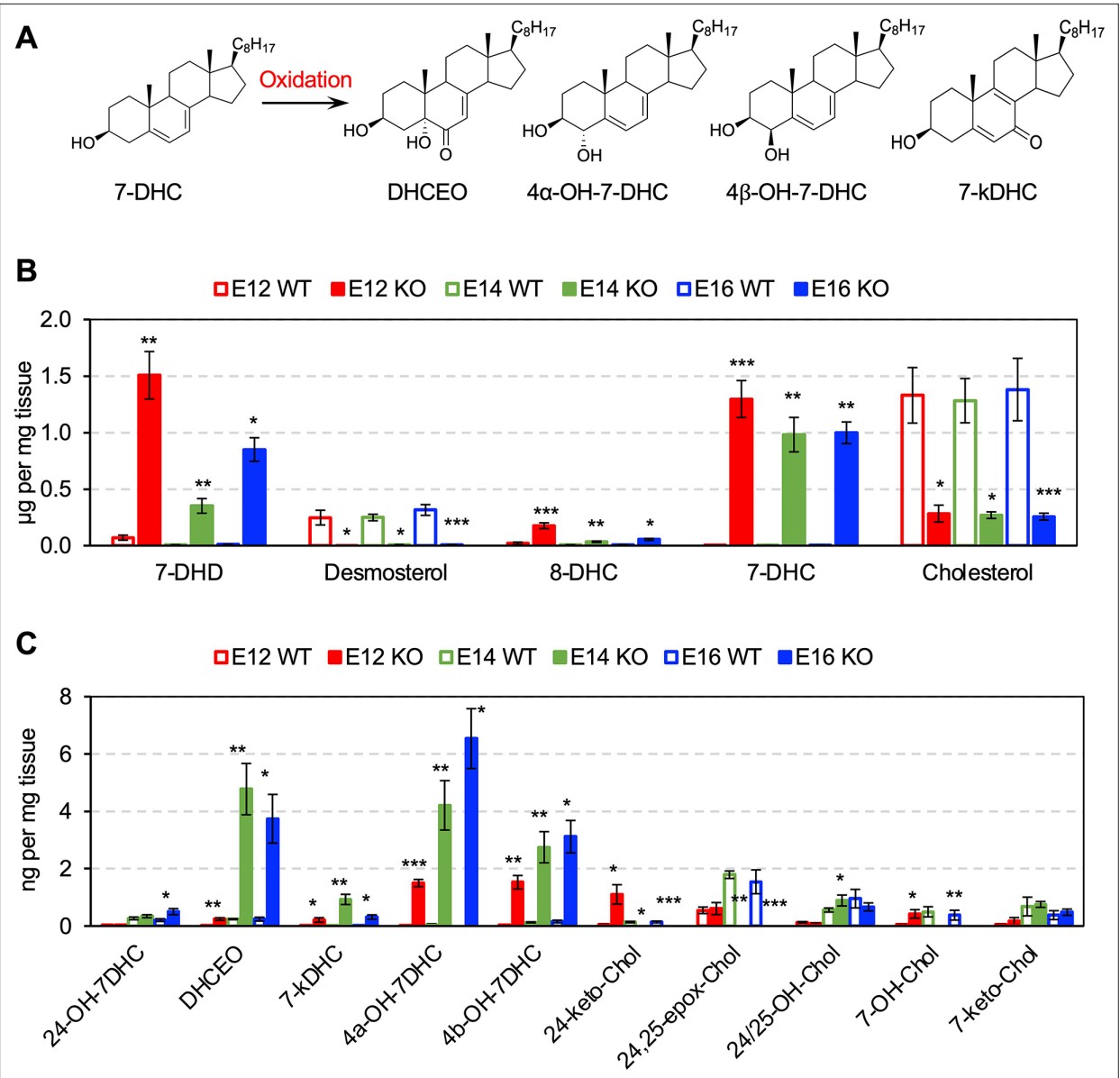

**Figure 2.** Cholesterol precursor 7-DHC and 7-DHC-derived Oxysterols are accumulated in *Dhcr7⁻/⁻* mouse embryonic cortex. (**A**) Chemical structures of 7-DHC-derived oxysterols. LC-MS/MS analysis of (**B**) cholesterol and its precursors and (**C**) 7-DHC and cholesterol-derived oxysterols in *Dhcr7⁺/⁺* and *Dhcr7⁻/⁻* embryonic cortex during development. Error bars indicate standard deviation. *, p<0.05; **, p<0.005; ***, p<0.001; n=3 biological replicates per group.

The online version of this article includes the following figure supplement(s) for figure 2:

**Figure supplement 1.** Cholesterol biosynthesis pathway, the enzymes involved in the synthesis and the diseases associated with the defective enzymes.

**Figure supplement 2.** Cholesterol precursor 7-DHC and 7-DHC-derived Oxysterols are accumulated *in* SLOS-derived human iPSCs and NPCs.

chromatography-tandem mass spectrometry (LC-MS/MS) was performed on cortices from *Dhcr7⁻/⁻* and *Dhcr7⁺/⁺* embryos during cortical development. These analyses revealed significant accumulation of 7-DHC and reduction of cholesterol in the embryonic cortices from *Dhcr7⁻/⁻* embryos throughout cortical development from E12.5 to E16.5 (*Figure 2B*). Increased levels of 7-dehydrodesmosterol (7-DHD), the precursor to desmosterol via Dhcr7, and 8-dehydrocholesterol (8-DHC) and decreased levels of desmosterol in *Dhcr7⁻/⁻* cortices are also consistent with the loss of the function of Dhcr7. Furthermore, 7-DHC oxysterols, such as DHCEO, 4α-hydroxy-7-DHC (4α-OH-7DHC), and 4β-hydroxy-7-DHC (4β-OH-7DHC) (*Xu et al., 2013*; *Xu et al., 2011*) showed substantial accumulation in the *Dhcr7⁻/⁻* cortices, reaching up to 4–8 ng/mg of tissue, throughout the development (*Figure 2A and C*,

and *Figure 2—figure supplement 1*). Note that 4 ng/mg of tissue would translate into approximately 10 µM assuming the density of the brain is 1 g/mL. On the other hand, cholesterol-derived oxysterols, such as 24-keto-Chol, 24,25-epoxy-Chol, 24- or 25-OH-Chol, 7-OH-Chol, and 7-keto-Chol decreased or did not change significantly in *Dhcr7*$^{-/-}$ cortices relative to *Dhcr7*$^{+/+}$ cortices.

Furthermore, significant accumulation of 7-DHC and 7-DHC-derived oxysterols and reduction of cholesterol were also found in both SLOS patient-derived hiPSCs and NPCs relative to Controls (*Figure 2—figure supplement 2*). Taken together, *DHCR7* mutations led to the accumulation of 7-DHC and 7-DHC-derived oxysterols in murine and human NPCs.

## Knockdown (KD) of Dhcr7 causes increased neurogenesis and depletion of cycling precursors in murine and human NPC cultures

To examine the potential role of *Dhcr7*, we generated three *Dhcr7* short hairpin RNAs (shRNAs)-EGFP reporters and transfected them into 293T cells along with murine *Dhcr7* cDNA-expressing plasmids. We found that *Dhcr7* shRNA2 was the most effective among those shRNAs and was chosen to examine *Dhcr7* function during neurogenesis from E12.5 cortical precursors in culture (*Figure 3A*). When cortical precursors were transfected with *Dhcr7* shRNA2-EGFP, the shRNA significantly decreased the percentage of transfected cells expressing *Dhcr7* (*Figure 3B–C*). *Dhcr7* KD led to a significant decrease in the proliferation of cortical precursors as measured by Ki67 immunoreactivity 3 days post-transfection or by adding 5-ethynyl-2-deoxyuridine (EdU) to culture 1 day post-transfection and immunostaining for EdU 2 days later (*Figure 3D–F*). The significant decrease in precursor proliferation was caused by a decrease in Pax6 +radial precursors (*Figure 3G*). Furthermore, *Dhcr7* KD led to a significant increase of βIII-tubulin+neurons 3 days post-transfection, but it did not affect cell survival as examined by immunostaining for CC3 (*Figure 3D, H and I*).

*Dhcr7* KD also decreased self-renewal of the radial precursor as demonstrated by clonal analysis with piggyBac (PB) transposon, which permanently labels precursors and their progeny. Cortical precursors were transfected with PB transposase and PB-*Dhcr7* shRNA2-EGFP or control shRNA-EGFP, and cultures were immunostained 3 days post-transfection for EGFP, the precursor markers, Sox2, and βIII-tubulin (*Figure 3J–M*). KD of *Dhcr7* reduced EGFP + multicellular clones (*Figure 3K*) whereas neuron-only (βIII-tubulin+) clones were increased (*Figure 3L*). Furthermore, the number of precursors in clones containing at least one Sox2 + cell is decreased (*Figure 3M*).

To ensure that these changes were *Dhcr7* shRNA-dependent, we performed rescue experiments using human *DHCR7* cDNA that is resistant to the murine *Dhcr7* shRNA (*Figure 3—figure supplement 1*). The murine shRNA did not affect the expression of human DHCR7 cDNA as confirmed by co-transfection of the murine shRNA and human *DHCR7* cDNA in 293T cells and western blot analysis 2 days later (*Figure 3—figure supplement 1A*). Furthermore, precursors were co-transfected with murine *Dhcr7* shRNA-EGFP or control shRNA-EGFP +/- human *DHCR7* cDNA and immunostained 3 days later for Ki67 or βIII-tubulin (*Figure 3—figure supplement 1B, C*). The human *DHCR7* cDNA showed significant rescue of the murine *Dhcr7* shRNA KD phenotypes, confirming that the neurogenesis and proliferation phenotype is *Dhcr7* shRNA-dependent.

Finally, we asked whether *DHCR7* also plays a role in human iPSC-derived NPCs (*Figure 3N–R*). To examine the function of DHCR7, we generated a human-targeted *DHCR7* shRNA-EGFP and confirmed its efficiency by transfecting it into 293T cells along with human DHCR7 expressing vectors (*Figure 3N*). Human NPCs were transfected with this shRNA-EGFP and analyzed by immunostaining 4 days later (*Figure 3O*), which showed that the KD of human *DHCR7* reduced EGFP + and Ki67 + proliferating precursors whereas increased EGFP + and βIII-tubulin+newborn neurons (*Figure 3P–R*), similar to those observed in murine cortical precursors and SLOS hNPCs.

## 7-DHC derived oxysterols lead to similar neurogenic defects as loss of Dhcr7 in murine cortical precursors in vitro

Oxysterols have been found to influence biological processes, including proliferation, differentiation, and cell survival (*DuSell et al., 2008*; *Theofilopoulos et al., 2013*; *Voisin et al., 2017*). Thus, we asked whether 7-DHC-derived oxysterols play regulatory roles in cortical precursor biology. To assess the potential effect of these 7-DHC-derived oxysterols on cortical precursor behaviors, cortical precursors from E12.5 *Dhcr7*$^{+/+}$ embryos were cultured and treated with different concentrations of individual oxysterols for 3 days, followed by immunostaining for Ki67, βIII-tubulin, and CC3

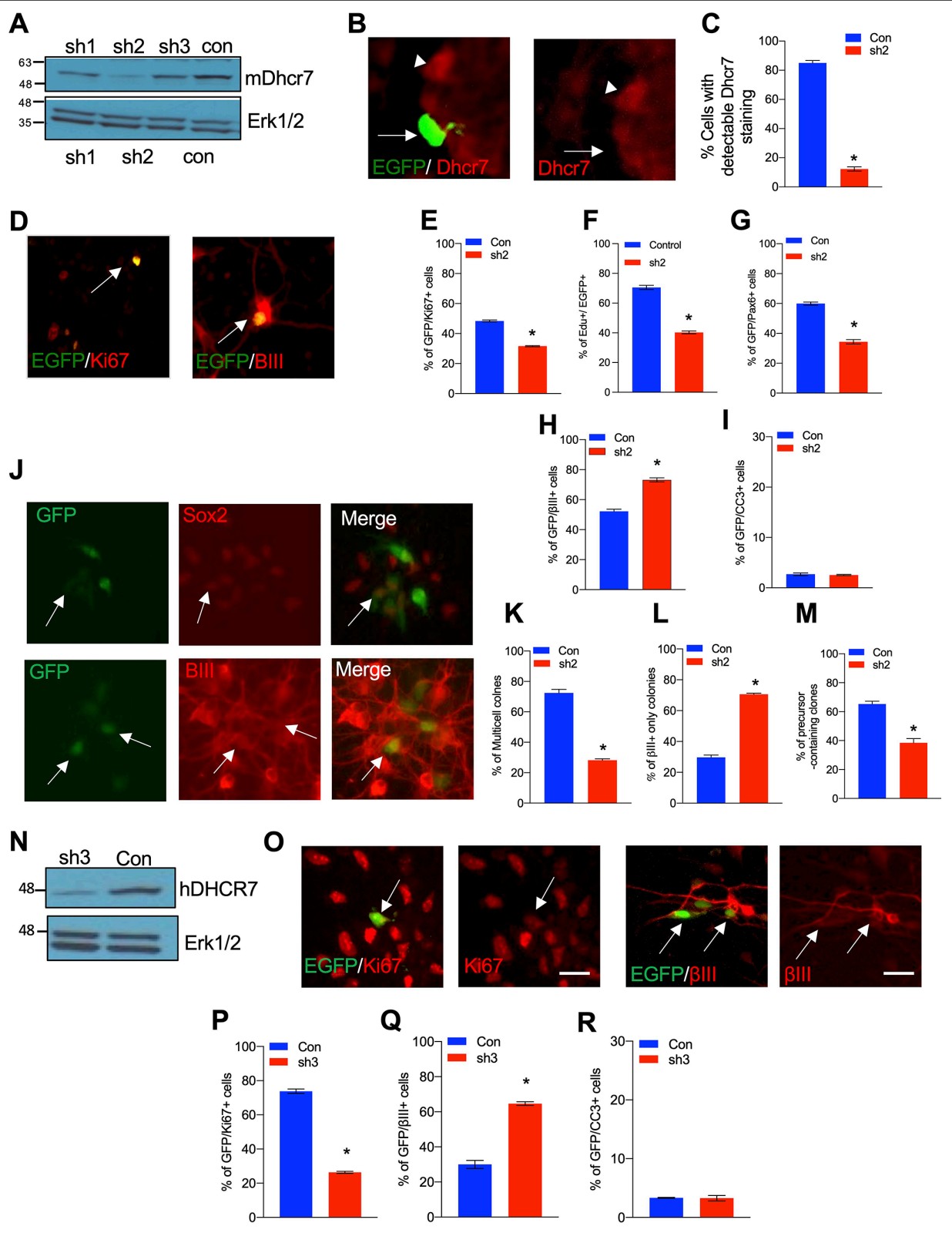

**Figure 3.** Knockdown of *Dhcr7* causes increased neurogenesis and depletion of cycling precursors in murine and human NPC cultures. (**A**) Western blot for Dhcr7 in 293T cells transfected with control or individual murine *Dhcr7* shRNAs. The blot was re-probed for Erk1/2 as a loading control. (**B and C**) Mouse cortical precursors were transfected EGFP-*Dhcr7* shRNAs (sh2) or EGFP-control (Con) and immunostained for EGFP and Dhcr7 (red) 2 days later and EGFP + cells expressing detectable Dhcr7 were quantified by fluorescence intensity (**C**). Arrow and arrowhead in (**B**) denote EGFP+/Dhcr7- and

*Figure 3 continued on next page*

*Figure 3 continued*

EGFP-/Dhcr7 + cells, respectively. (**D–G**) E12.5 cortical precursors were transfected with control or *Dhcr7* shRNAs and analyzed 3 days later. (**D**) Cultures were immunostained for EGFP (green) and Ki67 or βIII-tubulin (red; double-labelled cells in orange are indicated with arrows) or CC3. (**E–I**) The proportion of total EGFP + cells that were also positive for Ki67 (**E**), EdU 2 days after labeling (**F**), Pax6 (**G**), βIII-tubulin (**H**) or CC3 (**I**) was quantified. *, p<0.001; n=3. Scale bar = 50 µm. In all cases, error bars denote SEM. (**J–M**) E12.5 precursors were co-transfected with the PB transposase and PB-EGFP-control (Con) or PB-EGFP-*DHCR7* shRNA (sh2). (**J**) Cultured cells were immunostained for EGFP (green), Sox2 (red) and βIII-tubulin (red) after 3 days and quantified for clones greater than one cell in size (**K**), neuron-only clones (**L**), and clones with at least one Sox2 + precursors (**M**). Arrows in (**J**) top denote EFGP +/ Sox2 + precursors. Arrows in (**J**) bottom denote EGFP+/βIII-tubulin + cells. *, p<0.001; n=3. (**N**) Western blots of DHCR7 in 293T cells transfected with human control (Con) or human-specific *DHCR7* shRNA (sh3) plus human *DHCR7*-expressing plasmid, analyzed after 24 hr. The blot was re-probed for Erk1/2. (**O**) Human cortical precursors were transfected with EGFP-control (Con) or EGFP-*DHCR7* shRNA (sh3). Cells were immunostained 3 days later for EGFP (green) and Ki67 (red), βIII-tubulin (red) or CC3 and the proportion of total EGFP + cells that were also positive for Ki67 (**P**), βIII-tubulin (**Q**), or CC3 (**R**) was quantified. *, p<0.001; n=3. Scale Bar = 50 µm. Allows in (**N**) denote double-positive cells.

The online version of this article includes the following source data and figure supplement(s) for figure 3:

**Source data 1.** Related to *Figure 3A*.

**Source data 2.** Related to *Figure 3N*.

**Figure supplement 1.** Rescue of the neurogenesis phenotype in *Dhcr7*-knockdown mouse cortical precursors by human *DHCR7* cDNA expression vector.

(*Figure 4A*). Notably, DHCEO-treated cortical precursors showed a significant increase in the proportion of βIII-tubulin+newborn neurons and a significant decrease in the proportion of Ki67 + precursors (*Figure 4B–C*). This effect of DHCEO was observed in a dose-dependent manner up to 3.5 µM, which is roughly the concentration of DHCEO in the brains of P0 *Dhcr7*−/− mice (*Xu et al., 2011*). There was no significant change in cell death/survival with DHCEO treatment (*Figure 4D*). The other oxysterols showed a notable toxic effect at a concentration of 5 µM as indicated by increased CC3 + cells (*Figure 4G, J and M*). Note that the level of 7k-DHC is the lowest among the oxysterols, only reaching 0.9 ng/mg of tissue (equivalent to 2.3 µM), so we do not expect the highest concentration examined, 5 µM, for this oxysterol is relevant to the neurogenic phenotype. Interestingly, 4α-OH-7DHC treatment at a concentration of 2 µM also displayed significant increases in βIII-tubulin+neurons and decreases in Ki67 + precursors without significant changes in cell death/survival (*Figure 4H–J*). However, at 5 µM, CC3 + cells were significantly elevated at the expense of βIII-tubulin+neurons (*Figure 4J*). To summarize, treatment with DHCEO at physiologically relevant concentrations completely replicated the premature neurogenic phenotype observed in murine and human SLOS NPCs while 4α-OH-7DHC may also contribute to the phenotype despite its toxicity at high concentrations.

## Loss of Dhcr7 causes premature differentiation and reduced thickness of cortical layers in vivo

We asked whether loss of *Dhcr7* disrupted cortical development in vivo as was observed in vitro. To examine this, cortical sections from *Dhcr7*−/− mice at E18.5 were immunostained for the expression of well-defined cortical layer markers: the layer 2–4 marker Brn2, the layer 5 marker CTIP2, and the layer 6 marker Tbr1 (*Molyneaux et al., 2007*; *Figure 5A*). The analyses showed loss of *Dhcr7* led to a reduction of all cortical layers, resulting in the reduction of the overall size of the neocortex (*Figure 5B, C*). To further evaluate the effects of *Dhcr7*−/− in the fate decision of cortical precursors during corticogenesis, we immunostained cortical sections from *Dhcr7*−/− mice at E14.5 and E15.5 for Tbr1 and Satb2, which label early-born and late-born cortical neurons, respectively (*Figure 5D–E*). The analyses showed a significant increase of Tbr1 + neurons and Satb2 + neurons in the cortical section from *Dhcr7*−/− mice at E14.5 and E15.5, respectively (*Figure 5F–G*). This aberrant increased production of neurons with loss of *Dhcr7* potentially alters the number and localization of developing neurons in the cortex. To test this hypothesis, we injected pregnant dams with EdU at E12.5 to label proliferating radial precursors and analyzed *Dhcr7*+/+ and *Dhcr7*−/− littermates 3 days later at E15.5 to identify the locations of EdU-labeled cells in the cortex (*Figure 5H–I*). The majority of EdU-labeled cells migrated into the cortical plate in the KO mice, whereas EdU-labeled cells were scattered throughout the intermediate zone (IZ) and the cortical plates in the *Dhcr7*+/+ littermates. These observations suggest premature increases in precursor differentiation, which raises the possibility of activating major neurogenic signaling pathways during corticogenesis. RTKs and their downstream targets are known to play important roles in precursor proliferation and differentiation in the developing cortex, and TrkB,

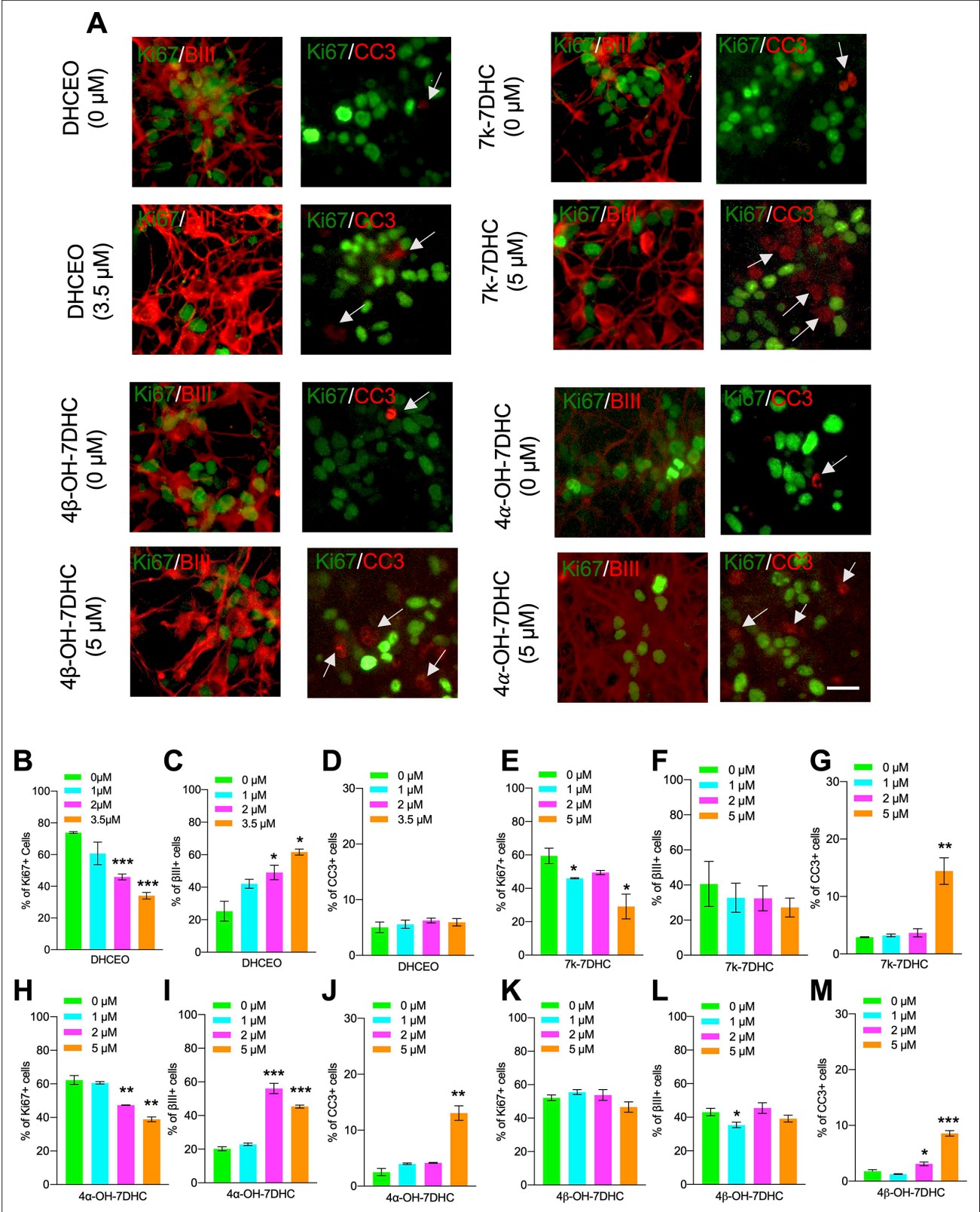

**Figure 4.** 7-DHC derived oxysterols lead to similar neurogenic defects as loss of *Dhcr7* in murine cortical precursors in vitro. (**A–M**) E12.5 cortical precursors were cultured for 2 days in the presence of different concentrations of 7-DHC-derived oxysterols and quantified. (**A**) Cell were immunostained for Ki67 (green), βIII-tubulin (red), and CC3 (red, arrow) after 3 days and the proportions of Ki67+ (**B,E,H,K**), βIII-tubulin+ (**C,F,I,L**), and CC3+ (**D,G,J,M**) cells were determined. Error bars indicate SEM. *, p<0.05; **, p<0.005; ***, p<0.001. n=3 per experiment. Scale Bar = 50 μm.

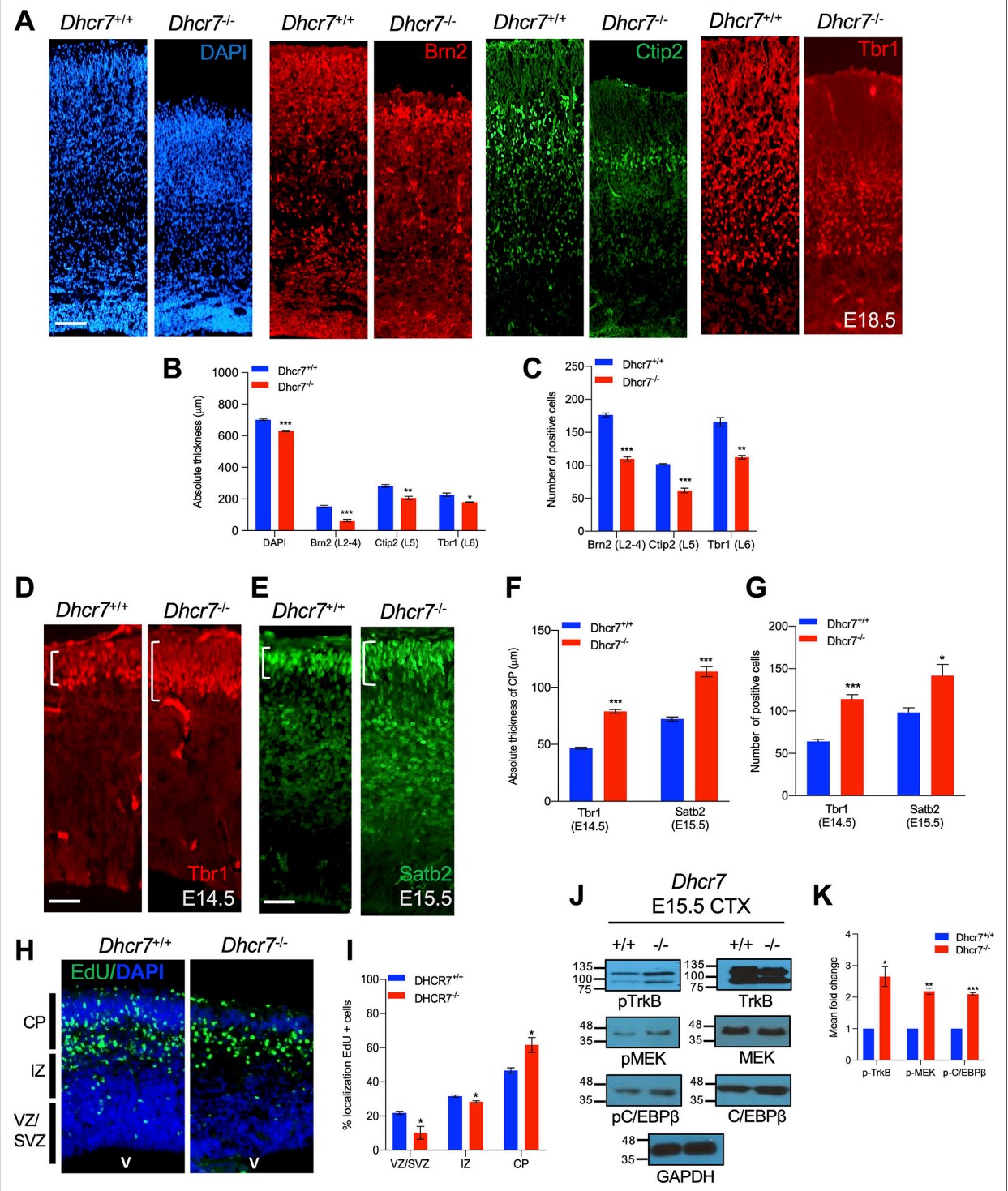

**Figure 5.** *Dhcr7*⁻/⁻ mice display premature neurogenesis and increased activity of the TrkB neurogenic signaling pathway in vivo. (**A**) E18.5 cortical sections from *Dhcr7*⁺/⁺ and *Dhcr7*⁻/⁻ were immunostained for Tbr1 (red), Ctip2 (green) and counterstained with DAPI (blue). (**B and C**) Quantifications of the absolute thickness (**B**) and the number of positive cells (**C**). (**D–G**) Cortical sections from E15.5 *Dhcr7*⁺/⁺ and *Dhcr7*⁻/⁻ mice were immunostained for Satb2 (D, green) and Tbr1 (E, red). Quantifications of the absolute thickness (**F**) and the number of positive cells (**G**) for Satb2 and Tbr1. (**H**) E15.5 cortical sections from *Dhcr7*⁺/⁺ and *Dhcr7*⁻/⁻ embryos EdU-labeled at E12.5 were immunostained for EdU (green) and counterstained with DAPI (blue). (**I**) Quantification of the relative location of EdU + cells in cortical sections. (**J**) E15.5 cortices were isolated from *Dhcr7*⁺/⁺ and *Dhcr7*⁻/⁻ embryos and analyzed by western blot for phospho-TrkB, phospho-MEK, or phospho-C/EBPβ. Blots were re-probed with antibodies for total GR, TrkB, MEK, C/EBPβ and GAPDH as loading controls. (**K**) Quantification of phospho-TrkB, phospho-MEK, and phospho-C/EBPβ expression in E15.5 cortices were isolated

*Figure 5 continued on next page*

Figure 5 continued

from *Dhcr7*[+/+] and *Dhcr7*[-/-] embryos. The relative levels of the phosphorylated proteins are normalized to GAPDH levels for each independent sample and expressed as fold increase. Error bars indicate SEM. *, p<0.05; **, p<0.005; ***, p<0.001. n=3 per experiment. Scale Bar = 50 μm.

The online version of this article includes the following source data for figure 5:

**Source data 1.** Related to *Figure 5J*.

one of the major RTKs expressed in cortical precursors, regulates proliferation and differentiation into neurons by activating the MEK-ERK-C/EBP pathway (*Barnabé-Heider and Miller, 2003*; *Bartkowska et al., 2007*; *Ménard et al., 2002*). Thus, we examined the activity of TrkB-MEK-ERK-C/EBP pathway by western blot with antibodies for phosphorylated activated MEK1 and C/EBPβ at E15.5 of *Dhcr7*[-/-] cortices (*Figure 5J–K*). This analysis showed that relative to total TrkB, MEK1, and C/EBPβ levels, their phosphorylated forms increased in *Dhcr7*[-/-] cortices (*Figure 5K*). This observation suggests that loss of *Dhcr7* can dysregulate an RTK-dependent signaling pathway that is critical for neural precursor proliferation and differentiation.

The above observations indicate premature precursor differentiation at the expense of depleting the proliferative precursor pool. To examine this hypothesis, pregnant dams were injected with Edu at different time points of cortical development, and *Dhcr7*[+/+] and *Dhcr7*[-/-] littermates were analyzed 18 hr later to quantify the number of cycling progenitors that have exited the cell cycles (*Figure 6A, B*). Cells that had left the cell cycle were stained as Edu + and Ki67-, whereas cells that re-entered the cell cycle were stained as Edu + and Ki67+. The index of the cell cycle exit is defined by the percentage of Edu+/Ki67- cells in total Edu + cells. The index showed that the number of cells exiting the cell cycle in *Dhcr7*[-/-] cortices was significantly increased at E13.5 and E14.5 compared with *Dhcr7*[+/+] cortices, indicating that loss of *Dhcr7* increased cell cycle exit. Increased cell cycle exit of cortical precursors can potentially arise from changes in proliferation. To test this possibility, pregnant dams were pulsed with Edu for 2 hr to label all cells in S-phase and the fraction of cortical precursors in the S-phase, a proliferating index from E13.5 to E15.5 in *Dhcr7*[-/-] and *Dhcr7*[+/+] cortices, was quantified (*Figure 6C–D*). The proliferating index was determined by the percentage of Edu + and Ki67 + cells out of total Ki67 + cells, which provides an estimate of cell cycle length because the length of the S phase remains relatively constant in mammalian cells, whereas the length of the G1 phase regulates proliferation (*DiSalvo et al., 1995*). The analyses showed a significant decrease in the proliferating index in *Dhcr7*[-/-] cortices compared to *Dhcr7*[+/+] cortices from E13.5 to E15.5 (*Figure 6C–D*), indicating slower cycle progression and longer cycle length in *Dhcr7*[-/-] cortices (*Chenn and Walsh, 2002*). Indeed, this accelerated depletion of the progenitor pool in *Dhcr7*[-/-] cortices results in decreases in the size of VZ/SVZ compared to *Dhcr7*[+/+] cortices from E13.5 to E15.5 (*Figure 6E–F*). Taken together, these results suggest that loss of *Dhcr7* leads to a decrease in the overall number of cortical precursors with increased cell cycle exit and decreased proliferation index via activation of the RTK-mediated MEK-ERK-C/EBP pathway.

## Perturbation in DHCR7 causes gene expression changes in neurogenic pathways in human NPCs

To ask whether *DHCR7* mutations influence the NPC biology, we aim to compare gene expression profiles between hNPCs with *DHCR7* mutations versus hNPCs from healthy individuals. To do so, we expanded the hNPCs derived from hiPSCs in the presence of essential growth factors and carried out RNA sequencing (RNAseq) on these cells. Relative to Control hNPCs, 2357 genes were significantly differentially expressed in SLOS hNPCs (1072 upregulated and 1285 downregulated genes with adjusted p-value < 0.05), distinguishing the SLOS and Control precursors (*Figure 7A*; *Supplementary file 6* for RNAseq data). Gene ontology and pathway analysis found that the PI3K-Akt signaling pathway, MAPK signaling, and Ras signaling pathways are among those with the highest p-values (*Figure 7B–C*; *Supplementary file 5*). Notably, MAP2K1(also known as MEK1) was found to be upregulated in SLOS patient-derived hNPCs. MAP2K1 is a key signaling molecule in MAPK and Ras signaling pathways, which are known to play important roles in precursor proliferation and differentiation (*Bonni et al., 1999*; *Ménard et al., 2002*; *Yang et al., 2013*), consistent with the phenotypes that were observed in *Dhcr7*[-/-] mice and SLOS patient-derived hNPCs in vitro.

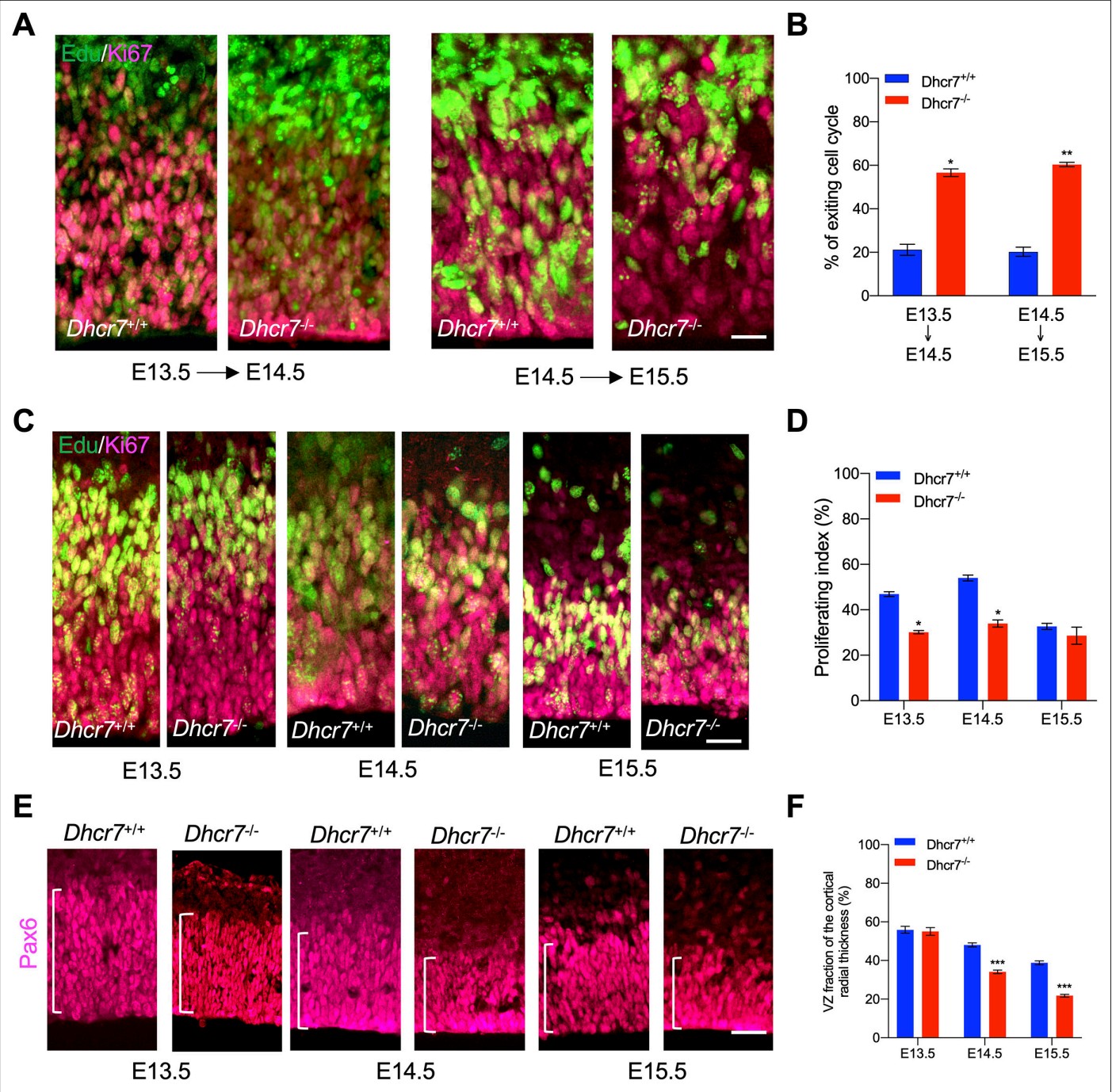

**Figure 6.** *Dhcr7⁻/⁻* mice show accelerated cell cycle exiting and depletion of cortical precursor cells in vivo. (**A**) Cortical sections from *Dhcr7⁻/⁻* and *Dhcr7⁺/⁺* embryos labeled by Edu at different developmental stages were immunostained 18 hr later for Edu (green) and Ki67 (Magenta). (**B**) Quantification of cell-cycle exit index of *Dhcr7⁻/⁻* cortices compared with *Dhcr7⁺/⁺* cortices. (**C**) Cortical sections from *Dhcr7⁻/⁻* and *Dhcr7⁺/⁺* embryos labeled by Edu injection at different developmental stages were immunostained 2 hr later for Edu (green) and Ki67 (Magenta). (**D**) Quantification of proliferation index of *Dhcr7⁻/⁻* cortices compared with *Dhcr7⁺/⁺* cortices. (**E**) Coronal cortical sections immunostained for Pax6 cortical precursor marker at different developmental stages. (**F**) Quantification of the relative size of the Pax6 +region shown as fractions of the whole cortical radial thickness. Error bars indicate SEM. *, p<0.05; **, p<0.005; ***, p<0.001. n=3 per experiment. Scale Bar = 50 µm.

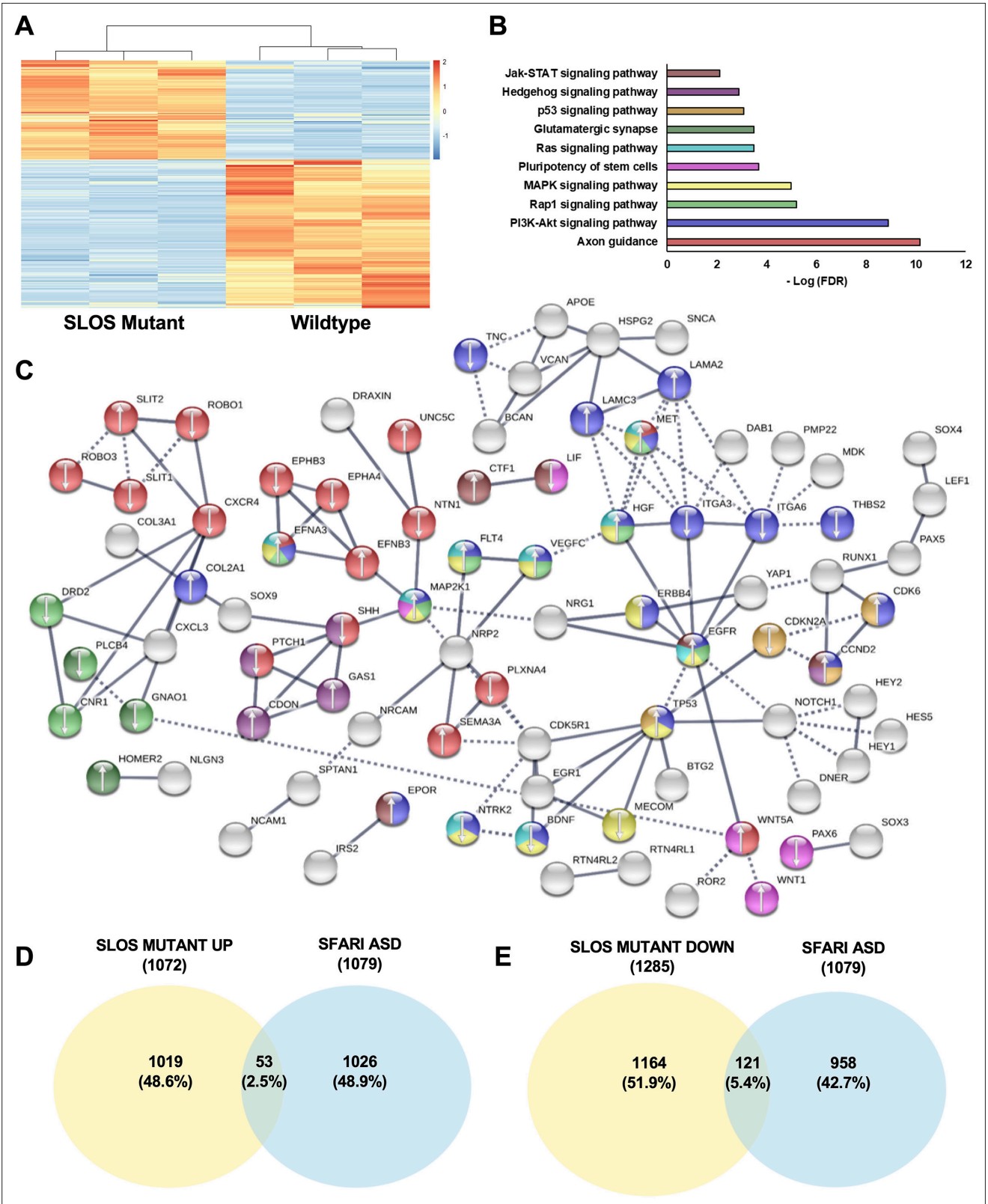

**Figure 7.** Loss of *Dhcr7* alters gene expression in cultured SLOS patient-derived neural progenitors. (**A**) Hierarchical clustering heatmap of differentially expressed genes shows distinct expression pattern changes in transcript abundance for SLOS mutant NPCs as compared to Controls. Red color represents an increase in abundance, blue color represents a relative decrease, and white color represents no change. (**B**) Enriched KEGG pathways identified among DEGs from SLOS mutant NPCs and involved in the Biological Function 'development of the central nervous system', identified by

*Figure 7 continued on next page*

*Figure 7 continued*

Ingenuity Pathway Analysis (see **Supplementary file 5**). Color of the bar corresponds to DEGs in the STRING network that are in enriched pathways. (**C**) String analysis of DEGs from SLOS mutant NPCs and involved in the Biological Function 'Development of the central nervous system' identified by Ingenuity Pathway Analysis. Parameters of high confidence have been applied and only connected nodes are displayed. Arrows indicate whether the gene was up- or down-regulated. Color of DEG corresponds to enriched KEGG pathway. (**D**) Venn diagram showing the overlap of autism risk genes in the SFARI database with genes upregulated in SLOS mutant NPCs. (**E**) Venn diagram showing the overlap of autism risk genes in the SFARI database with genes downregulated in SLOS mutant NPCs. n=3 biological replicates per genotype; DEGs met the criteria of fold-change >1.5 between genotypes and adjusted p-value < 0.05.

## 7-DHC-derived oxysterol, DHCEO, activates cortical neurogenesis via activation of the glucocorticoid receptor (GR) and inhibition of GR or inhibition of the formation of DHCEO rescues the neurogenic defects in SLOS NPCs

As shown in **Figure 4**, treatment of WT mouse NPCs with 7-DHC-derived oxysterols, particularly DHCEO, can replicate the same aberrant premature neurogenesis observed in SLOS NPCs. However, the mechanism of action of DHCEO remains unknown. Interestingly, 6-oxo-cholestan-3β,5α-diol (OCDO), a cholesterol-derived oxysterol that is structurally similar to DHCEO, is shown to bind and activate glucocorticoid receptor (GR) (**Voisin et al., 2017**; **Figure 8—figure supplement 1A**). Additionally, GR activation has been shown to activate TrkB (**Jeanneteau et al., 2008**), which leads to further activation of the RTK-mediated MEK-ERK pathway (**Barnabé-Heider and Miller, 2003**; **Bartkowska et al., 2007**; **Ménard et al., 2002**). TrkB and RTK-mediated MEK-ERK pathway are necessary for neurogenesis during embryonic cortical development. Thus, we asked whether DHCEO can activate the GR and further lead to the activation of the MEK-ERK neurogenic pathway. To investigate this, we first examined if GR was phosphorylated and activated in the embryonic cortices from $Dhcr7^{-/-}$ and $Dhcr7^{+/+}$ embryos at E15.5 in addition to TrkB, MEK, and C/EBPβ shown in **Figure 5J**. Western blots demonstrated that phosphorylation of GR was indeed increased in $Dhcr7^{-/-}$ cortices compared to $Dhcr7^{+/+}$ cortices (**Figure 8A**).

To ask if DHCEO could physically interact with GR, we performed a molecular docking simulation between DHCEO and the ligand binding domain of GR. Molecular docking is an effective computational approach to understand protein-ligand interaction between small molecules and receptor proteins both energetically and geometrically (**Ferreira et al., 2015**). As seen in **Figure 8B**, DHCEO docks in the binding pocket of human GR favorably with a docking score of –9 (a negative value suggests favorable interactions), comparable to the docking score of OCDO at –9.5. Additional docking positions show that many of the predicted positions for OCDO can be recapitulated with DHCEO (**Figure 8—figure supplement 1**).

We further examined whether DHCEO could initiate GR and TrkB activation in Control hNPCs in vitro. Cultured hNPCs were exposed to DHCEO (3.5 μM), a physiologically relevant concentration, harvested at the different time points (3, 6, 10, and 24 hr), and evaluated for GR and TrkB activation by western blots (**Figure 8C**), which revealed that DHCEO-treated hNPCs showed a gradual increase of phosphorylated GR starting at 3 hr and peaking at 10 hr of exposure. As GR phosphorylation increased, DHCEO-treated hNPCs also showed the phosphorylation of TrkB from 6 hrs to 10 hr exposure, which dissipated at 24 hr exposure. As no exogenous GR ligands and neurotrophins were added in these experiments, these results indicated that DHCEO caused the increased phosphorylation of GR, which led to the activation of TrkB phosphorylation in human NPCs in vitro.

To confirm that GR activation could cause aberrant premature neurogenesis, we cultured human cortical precursors and transfected them with $hDHCR7$ shRNA-EGFP and control shRNA-EGFP. Twelve hr post-transfection, a glucocorticoid receptor antagonist, RU38486, or vehicle control was added and cultured for 3 days. Cultures were immunostained for EGFP and βIII-tubulin to evaluate the effects on neurogenesis. The analysis showed that inhibition of GR by RU38486 rescued the $DHCR7$-KD-mediated increase in neurogenesis down to the level observed in the control shRNA (**Figure 8D**).

These data suggest that premature cortical neurogenesis is promoted by GR activation by DHCEO and subsequent increased activation of the RTK-mediated MEK-ERK pathway. To directly evaluate this hypothesis, we inhibited MEK in cultured NPCs using two well-characterized MEK inhibitors, PD98059 and trametinib. The efficiency of these compounds in murine NPCs has been demonstrated previously (**Tomita et al., 2020**; **Barnabé-Heider and Miller, 2003**). Indeed, western blots demonstrated

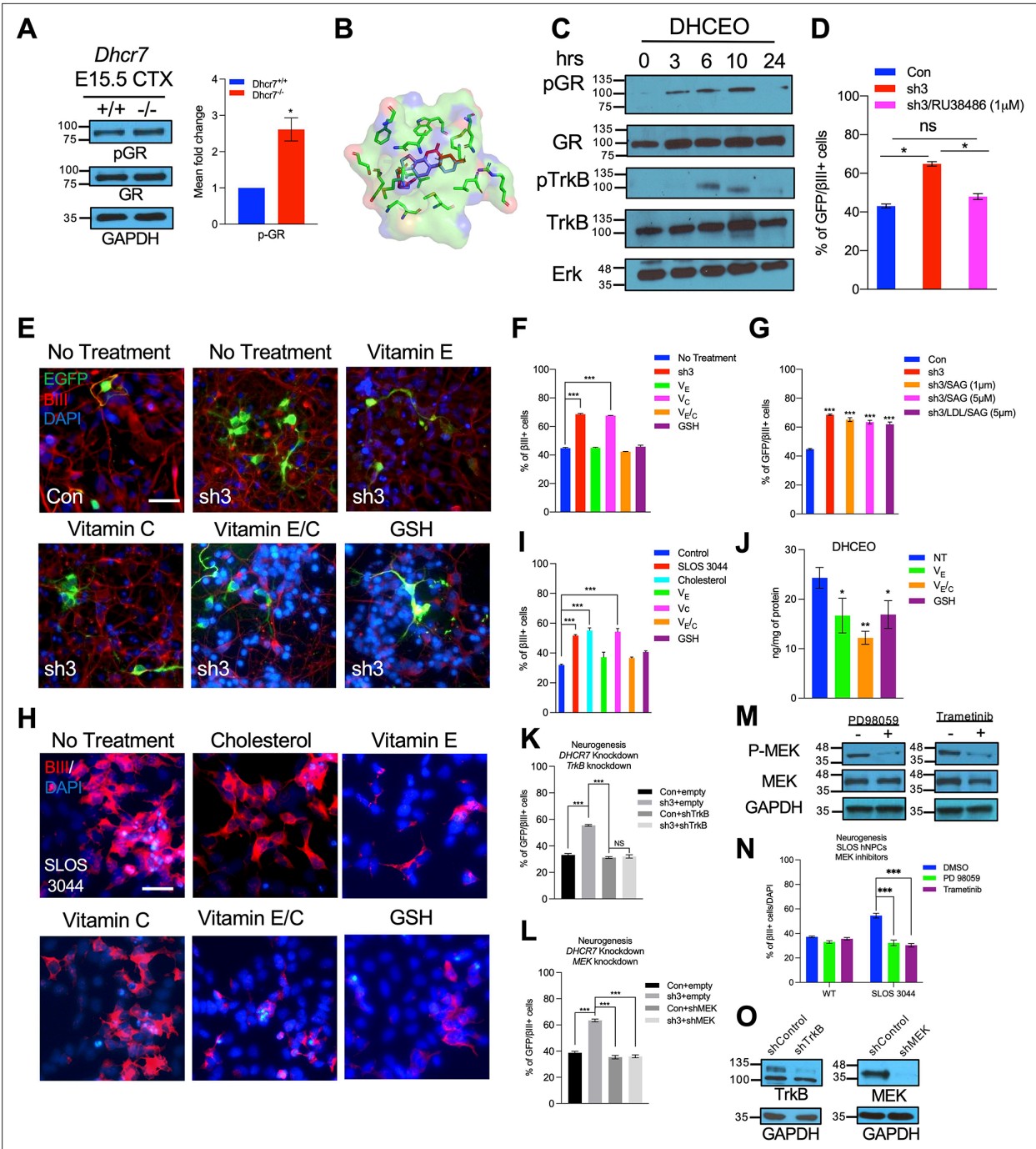

**Figure 8.** DHCEO activates cortical neurogenesis via activation of glucocorticoid receptor and inhibition of the effect or the formation of DHCEO rescues the neurogenic defects in SLOS NPCs. (**A**) Western blot showing increased phospho-GR in E15.5 *Dhcr7*[-/-] mouse brain relative to *Dhcr7*[+/+]. (**B**) Image of the docked position of DHCEO (red) and OCDO (blue) in the ligand binding pocket of GR. (**C**) Human neural progenitors were treated with 3.5 µM DHCEO over the indicated time periods. Lysates were probed with phosphor-GR and phosphor-TrkB and re-probed with antibodies for total GR, total TrkB or total ERK as loading controls. (**D**) Human NPCs were transfected with EGFP-control (Con) or EGFP-*DHCR7* (sh3) shRNA. Cells were treated with 1 µM RU38486, a selective GR antagonist 1 day after transfection. Three days post-transfection, cells were immunostained for EGFP and βIII-tubulin and quantified. (**E**) Control hNPCs were transfected with EGFP-control (Con) or EGFP-*DHCR7* shRNA (sh3), and then treated with vitamin E, vitamin C, vitamin E/C, or glutathione (GSH). Three days post-transfection, cells were immunostained for EGFP and βIII-tubulin. (**F,G**) Quantification of EGFP and βIII-tubulin + cells in *DHCR7*-KD Control hNPCs treated with various antioxidants, SAG, or LDL + SAG. (**H**) SLOS hNPCs were treated with cholesterol or various antioxidants and were immunostained for βIII-tubulin and DAPI. (**I**) Quantification of the proportion of βIII-tubulin + cells in Control hNPCs, and SLOS hNPCs treated with cholesterol (LDL) or various antioxidants. (**J**) Quantification of DHCEO by LC-MS/MS in SLOS hiPSCs treated with vitamin E, vitamin E/C, and GSH. (**K, L**) hNPCs were transfected with EGFP-control (Con) or EGFP-*DHCR7* and co-transfected with *TrkB* shRNA (**K**) or

*Figure 8 continued on next page*

*Figure 8 continued*

*MEK* shRNA (**L**) vector. Three days later, cultures were immunostained for EGFP and βIII-tubulin and the proportion of transfected newborn neurons was determined. (**M**) hNPCs were treated or not treated with MEK/ERK inhibitors, trametinib or PD98059. Western blot of phosphor-MEK. The blots were then re-probed with antibodies for total MEK as loading controls. (**N**) SLOS hNPCs were treated with vehicle control or MEK/ERK inhibitors, trametinib (100 nM; purple column) or PD98059 (50 μM; green column). Three days later, cultures were immunostained for βIII-tubulin, and the proportion of new neurons was determined. (**O**) Western blots for TrkB or MEK1/2 in the lysate of 293T cells transfected with control or *TrkB* shRNA or *MEK* shRNA vector. The blots were re-probed for glyceraldehyde 3-phosphate dehydrogenase (GAPDH). Error bars indicate SEM. *, p<0.05; **, p<0.005; ***, p<0.001. n=3 per experiment. Scale Bar = 50 μm.

The online version of this article includes the following source data and figure supplement(s) for figure 8:

**Source data 1.** Related to *Figure 8A*.

**Source data 2.** Related to *Figure 8C*.

**Source data 3.** Related to *Figure 8M*.

**Source data 4.** Related to *Figure 8O*.

**Figure supplement 1.** Full docking results of DHCEO and OCDO in the binding pocket of human glucocorticoid receptor.

**Figure supplement 2.** Antioxidant and MEK inhibitors rescued the neurogenic phenotype in human and murine NPCs with *Dhcr7* mutations.

**Figure supplement 3.** Cholesterol depletion did not disrupt the neurogenesis in human and mouse cultured neural precursors.

that both PD98059 and trametinib efficiently inhibited MEK activation in hNPCs (*Figure 8M*). Having confirmed the efficacy of these two compounds, we cultured SLOS hNPCs (SLOS 3044 and 5844) and wildtype hNPCs as well as murine *Dhcr7$^{-/-}$* NPCs and wildtype NPCs, and treated them with either PD58059 or trametinib. The cultures were immunostained for βIII-tubulin 3 days later to evaluate the effect of the inhibitors on neurogenesis. As expected, both SLOS hNPCs and murine *Dhcr7$^{-/-}$* NPCs increased the proportion of βIII$^+$ newborn neurons, and the MEK inhibition by either PD98058 or trametinib reversed increased neurogenesis with *DHCR7* mutations back to the level seen in the wild-type NPCs (*Figure 8N*, *Figure 8—figure supplement 2C, F*).

Chemical inhibitors may have off-target effects. Therefore, we performed rescue experiments by genetically knockdown either TrkB or MEK. We evaluated the efficiency of shRNA targeting TrkB and MEK by transfecting them independently into 293T cells and performed western blots (*Figure 8O*). These analyses showed that TrkB or MEK shRNA decreased TrkB or MEK protein whereas control shRNA did not. Either TrkB or MEK shRNA was transfected to cultured human neural progenitors with EGFP-*DHCR7* shRNA. Quantification of EGFP$^+$, βIII-tubulin + cells 3 days later demonstrated that coincident TrkB knockdown rescued the increased neurogenesis phenotype observed following shRNA-mediated *DHCR7* knockdown (*Figure 8K*). Similarly, MEK knockdown also reversed the premature neurogenesis with *DHCR7* shRNA knockdown in human NPCs (*Figure 8L*). Thus, *DHCR7* mutations disrupt the normal cortical neurogenesis by hyperactivating RTKs, such as TrkB, through GR activation and thus activating the downstream MEK-ERK1/2 pathway.

Finally, we assessed whether aberrant premature neurogenesis was due to DHCEO, as it is a major oxysterol derived from free radical (non-enzymatic) oxidation of 7-DHC (*Xu et al., 2011*; *Figure 2*). Antioxidants can prevent or slow down the formation of oxysterols via the free radical mechanism and protect cells from the deleterious effects of oxysterols (*Yin et al., 2011*). In particular, vitamin E (V$_E$), but not vitamin C (V$_C$), has been shown to effectively inhibit the formation of 7-DHC-derived oxysterols in SLOS patient-derived fibroblasts (*Korade et al., 2014*). Combination with V$_C$, which recycle oxidized V$_E$, enhances the antioxidant activity of V$_E$ against lipid peroxidation (*Doba et al., 1985*). Furthermore, glutathione (GSH) is an important detoxifying molecule that is abundant (in mM) in cells and can potentially inhibit the formation of DHCEO by reacting with its electrophilic precursor, 7-DHC 5α,6α-epoxide (*Porter et al., 2020*; *Xu et al., 2011*). To test this possibility, we treated *hDHCR7* KD human NPCs with antioxidants (V$_E$, V$_C$, V$_E$/V$_C$, or GSH) and then immunostained for EGFP and βIII-tubulin to evaluate the effects on neurogenesis (*Figure 8E–F*). The analysis indicated that V$_E$, V$_E$/V$_C$, or GSH, but not V$_C$ alone, effectively reversed increased neurogenesis caused by *hDHCR7* KD to the level observed in the control. Potential detoxifying effects of these antioxidants were further tested in human SLOS patient-derived NPCs as well as cortical precursors from *Dhcr7$^{-/-}$* mouse embryos in vitro (*Figure 8H–I* and *Figure 8—figure supplement 2A, B, D, E*). Treatments with V$_E$, V$_E$/ V$_C$ and GSH were again found to effectively reduce the increased neurogenesis in both human and murine NPCs with *Dhcr7* mutations down to the level observed in WT or Control samples. LC-MS/MS analysis

confirmed that $V_E$, $V_E/V_C$, or GSH indeed inhibited the levels of DHCEO in SLOS hiPSCs (*Figure 8J*). These results suggest that free radical chain-breaking antioxidants, such as $V_E$, and nucleophilic antioxidants, such as GSH, can effectively rescue the neurogenesis phenotype observed in SLOS NPCs by inhibiting the free radical oxidation of 7-DHC at different stages (*Figure 8—figure supplement 2G*).

In addition to antioxidant treatment, we evaluated whether cholesterol plays significant roles in normal neurogenesis in human and murine NPCs. We tested whether cholesterol depletion could phenocopy the premature neurogenesis observed in both human and murine DHCR7 mutations. Human and murine NPCs were treated with of β-cyclodextrin, a cholesterol-depleting agent, and the changes in the proportion of βIII-tubulin+neurons were quantified 3 days after plating (*Figure 8—figure supplement 3B*). LC-MS/MS analysis demonstrated that β-cyclodextrin treatment reduced cellular cholesterol by more than 50% (*Figure 8—figure supplement 3A*); however, it did not change the proportion of βIII-tubulin+neurons and Ki67 + progenitors either in human NPCs or murine NPCs (*Figure 8—figure supplement 3C, D*). Furthermore, we tested whether restoration of cholesterol could potentially rescue aberrant premature neurogenesis observed in both human and murine SLOS NPCs (*Figure 8H–I*, *Figure 8—figure supplement 2B, E*). Low-density lipoprotein was added to cultures to examine the effects of cholesterol on neurogenesis. The results revealed that addition of cholesterol did not prevent increased neurogenesis in SLOS NPCs (*Figure 8H–I*, *Figure 8—figure supplement 2B, E*). Because DHCEO has been shown to inhibit Smo in the Hh signaling pathway (*Sever et al., 2016*), we also tested whether supplementation of SAG (a Smo agonist) can rescue the phenotype with or without cholesterol supplementation in *DHCR7*-KD hNPCs (*Figure 8G*). However, none rescued the neurogenesis phenotype. Thus, the lack of cholesterol and inhibition of Hh signaling do not contribute to the neurogenesis phenotype observed in human and murine NPCs with *DHCR7* mutations.

Taken together, the abnormal premature neurogenesis in cortical precursor with *Dhcr7* mutations was caused by DHCEO activation of GR and further activation of Trk-mediated neurogenic signaling pathway, but not by the deficiency in cholesterol, and inhibition of the formation of DHCEO or GR rescues the neurogenic defects.

## Discussion

Whether deficiency in cholesterol, accumulation of 7-DHC, or accumulation of 7-DHC oxysterols is more important for the neurological and developmental phenotypes observed in SLOS patients remains unknown. Here, we characterized the neurogenic phenotype of SLOS cortical precursors in vitro and in vivo and showed that the accumulated 7-DHC-derived oxysterols, particularly DHCEO, disrupt normal embryonic neurogenesis by accelerating progressive depletion of cortical precursors through neurogenesis, ultimately resulting in greatly reduced cerebral cortex thickness with abnormal cortical layering.

The data presented here support four major conclusions. First, LC-MS/MS analyses indicate that *DHCR7* mutations promote a significant accumulation of 7-DHC-derived oxysterols in murine and human embryonic cortical NPCs during cortical development. Second, our studies with KD or KO of *DHCR7* indicate that *DHCR7* is necessary for the normal proliferation of embryonic NPCs and genesis of newborn neurons in culture and within the environment of the embryonic cortex. KD or KO of *DHCR7* leads to premature differentiation of cortical precursors to neurons and ultimately a reduction in the thickness of cortical layers. This occurs through the activation of the neurogenic TrkB-MEK-C/EBP pathway. Third, we show that 7-DHC-derived oxysterols, especially DHCEO, display detrimental effects on normal neural development, and that DHCEO can activate GR, which is responsible for abnormal cortical precursor differentiation, since *DHCR7* KD/KO phenotype can be rescued by concurrent inhibition of glucocorticoid receptor. Fourth, we show that premature differentiation of cortical precursors with *DHCR7* mutations is cholesterol-independent since inhibition of 7-DHC-derived oxysterols via antioxidant treatment can rescue *DHCR7* KD/KO phenotype, whereas cholesterol supplementation cannot.

Based on these data, we propose a model where DHCEO binds and activates GR, and in doing so, controls the RTK-mediated neurogenic pathway, TrkB-MEK-C/EBP, and thus fate-decision of developing precursors and neurons. Although docking studies only approximate enthalpic energies and are an incomplete approximation of affinity, they provide a structural rationale for this activity. This model is consistent with previous studies showing that an oxysterol structurally similar to DHCEO, OCDO,

can bind and regulate GR and its transcriptional activity (*Voisin et al., 2017*) and ligand-bound GR promoted TrkB phosphorylation, leading to activation of its downstream effectors, MEK and ERK (*Jeanneteau et al., 2008*). Our model of DHCEO exerting its effect on neurogenesis through GR does not contradict the previous finding by Francis et al., which suggests that Wnt/β-catenin inhibition contributes to precocious neurogenesis (*Francis et al., 2016*), because activation of GR has been shown to inhibit Wnt/β-catenin signaling in several systems (*Olkku and Mahonen, 2009*; *Zhou et al., 2020*).

This work represents the first comprehensive characterization of the neurogenic phenotype in the SLOS mouse model and SLOS patient-derived NPCs. We demonstrate that the aberrant neural development observed in mice and human NPCs with *DHCR7* mutations occurs in a cholesterol-independent and 7-DHC-derived oxysterol-dependent manner. Successful rescue of the neurogenic phenotype by inhibiting GR or inhibiting the formation of oxysterols with antioxidants pave the wave for potential therapies for the neurological defects observed in SLOS patients.

## Materials and methods

### Animals

All animal experiments were performed in accordance with the Guideline 'Guide for the Care and Use of Laboratory Animals' of the National Institutes of Health and were approved by the University of Washington Institutional Animal Care and Use Committee. *Dhcr7* mice (B6.129P2(Cg)-*Dhcr7*tm1Gst/J) were maintained as heterozygous and were genotyped as described previously (*Fitzky et al., 2001*). The primers used are the following: *Dhcr7*-WT-F: 5'-GGATCTTCTGAGGGCAGCTT-3'; *Dhcr7*-WT-R: 5'-TCTGAACCCTTGGCTGATC-3'; Delta-Mut: 5'-CTAGACCGCGGCTAGAGAAT-3'. For in vitro transfection, C57BL/6 J E12.5 pregnant mice were obtained by time-mating. Mating pairs of wild-type (*Dhcr7*+/+) C57BL/6 J mice were purchased from Jackson Laboratories (Bar Harbor, ME). All mice had free access to rodent chow and water in a 12 hr dark-light cycle room.

### Cells

Primary human fibroblasts isolated from SLOS patients (GM03044 and GM05788) were purchased from Coriell Institute. These lines have been sequenced by Coriell Institute to confirm their mutations that affect the function of DHCR7, and the biochemical defects have been confirmed by measuring the 7-dehydrocholesterol/cholesterol ratio. The cultures were also tested to be free of mycoplasma, bacteria, and fungi contamination by the vendor. Human embryonic kidney 293T cells were purchased from ATCC (catalog #: CRL-3216), which carried out authentication using STR profiling and tested to be mycoplasma free.

### Plasmids

The target sequence for murine *Dhcr7* shRNA was cloned into pLKO-UBI-GFP digested with EcoRI and PacI. pLKO-UBI-GFP was generated by digesting out hPGK promoter from pLKO.3G and ligating ubiquitin promoter (UBI) from pCLX-UBI-VenusN with PacI and BamHI. pLKO.3G was a gift from Christophe Benoist & Diane Mathis (Addgene plasmid # 14748; http://n2t.net/addgene:14748; RRID:Addgene_14748) and pCLX-UBI-VenusN was a gift from Patrick Salmon (Addgene plasmid # 27247; http://n2t.net/addgene:27247; RRID:Addgene_27247). cDNAs encoding murine (pCMV-mDHCR7-Myc/DDK) and human DHCR7 (pCMV-hDHCR7-Myc/DDK) were purchased from Origene (MR223420 and RC228922 for murine and human cDNA clone respectively). PB-EF1α-GreenPuro-H1MCS and Super PiggyBac transposase expression vector from System Biosciences (Cat#PBS506A-1 and Cat# PB210PA-1 respectively) were used for clonal analysis.

### Cortical precursor cell cultures

Murine cortical precursor cells were cultured as described previously (*Barnabé-Heider and Miller, 2003*; *Tomita et al., 2020*). Briefly, mouse cortical precursor cells from cortices were dissected from E12.5 *Dhcr7*+/+ or *Dhcr7*-/- (KO) mouse embryos in ice-cold HBSS (Invitrogen) and transferred into cortical precursor medium. The cortical precursor medium consisted of Neurobasal medium (Invitrogen) with 500 µM L-glutamine (Invitrogen), 2% B27 supplement (Invitrogen), 1% penicillin-streptomycin (Invitrogen) and 40 ng/ml FGF2 (BD Biosciences). The dissected tissue was mechanically

triturated by a fire-polished glass pipette and plated onto 24-well plates coated with 2% laminin (BD Biosciences) and 1% poly-D-lysine (Sigma Aldrich). Plating density of the cortical precursors was 150,000 cells/well in 24-well plates for single embryo cultures and plasmid transfections.

### Human induced pluripotent stem cells (hiPSCs)

Human fibroblasts (GM03044 and GM05788; see above) isolated from patients with SLOS were reprogrammed as described elsewhere (*Okita et al., 2011*; *Yu et al., 2007*). Briefly, 1x10⁶ cells of human fibroblast cells at early passages were trypsinized (0.25% Trypsin/0.5 mM EDTA, Gibco) and electroporated with indicated episomal plasmids using Amaxa Basic Nucleofector Kit for primary mammalian fibroblasts, program A-24 (Lonza). 1.0 µg pSIN4-EF2-O2S and pSIN4-EF2-N2L were used for each electroporation. The electroporated human fibroblasts were seeded onto a gelatin-coated tissue culture dish and cultured with fibroblast medium (DMEM with 10% FBS, 2 mM L-glutamine, 1% non-essential amino acids, 1% sodium pyruvate, and 1% penicillin and streptomycin) for 7 days and re-seeded onto feeder layer cells (irradiated SNL 76/7 feeder cells; a gift from Dr. Allan Bradley, Sanger Institute, UK) with hiPSC medium (DMEM/F12 medium with 20% knockout serum replacer, 1% sodium pyruvate, 1% non-essential amino acids, 0.007% 2-mercaptoethanol,1% penicillin and strep-tomycin, 0.1 mM sodium butyrate, 50 nM suberoylanilide hydroxamic acid, and 4 ng/ml of bFGF) in 100 mm tissue culture dishes and continued to be cultured until hiPSC colonies were visible (28 days from electroporation). The hiPSC colonies were then picked and further cultured for expansion.

### Maintenance and differentiation of hiPSC

Undifferentiated hiPSCs were cultured and maintained in mTeSR Plus medium (StemCell Technologies) on Matrigel (BD Biosciences)-coated plates prior to the generation of human neural progenitor cells (hNPCs). hNPCs were generated using the STEMdiff SMADi Neural Induction kit (StemCell Technologies) according to the manufacturer's protocol. Briefly, neuralized embryoid bodies (EBs) were generated by culturing small aggregates of hESCs in ultra-low attachment plates (Corning) in STEMdiff SMADi Neural induction medium. EBs were replated and cultured in poly-D-lysine/Laminin-coated plates with the neural induction medium until EBs formed neural rosette formations. Neural rosettes were collected and replated in poly-D-lysine/Laminin-coated plates for hNPCs outgrowth in neural induction medium until hNPCs were ready for the first passage. hNPCs were maintained in STEMdiff Neural Progenitor medium. For neurogenesis of hNPCs, hNPCs were cultured in Neurobasal A medium supplemented with B27 minus vitamin A, 1% penicillin-streptomycin and glutamax (all from Invitrogen) in poly-D-lysine/Laminin-coated plates (*Konopka et al., 2012*; *Usui et al., 2017*). For quantification, an average of 700 cells was counted per condition in 5–6 random fields per independent experiment.

### In vitro differentiation of iPSCs

For in vitro differentiation of human iPSCs to the three germ layers, STEMdiff Trilineage Differentiation Kit (StemCell Technologies) was used as described by the manufacturer. Briefly, cells were harvested by Gentle Cell Dissociation Reagent (StemCell Technologies) and plated onto Matrigel-coated 24-well plates with mTeSR1 medium (StemCell Technologies). Cell density and viability were determined using trypan blue exclusion. Cells were seeded at a clonal density of 200,000 cells/cm² for ectoderm and endoderm differentiation and 50,000 cells/cm² for mesoderm differentiation. Twenty-four hr after plating, cells were switched to the appropriate STEMdiff Trilineage medium for ectoderm, mesoderm and endoderm differentiation. Cells were cultured in the lineage-specific medium for 5 days (meso-derm and endoderm lineages) and 7 days (ectoderm lineage) and were harvested and/or fixed for analyses of lineage-specific markers for the three germ layers (*Figure 1—figure supplement 1*).

### Transfection and quantification

For plasmid transfection of mouse cortical precursors, Lipofectamine LTX and Plus Reagent (Invit-rogen) were used as described by the manufacturer. Briefly, 1 µg of DNA and 1 µl of Lipofectamine LTX and Plus Regent in 100 µl of Opti-MEM (both from Invitrogen) were mixed, incubated for 20 min, and added to precursors three hours after plating. For clonal analysis, 1.5 µg DNA, at 1:3 ratio of Super PiggyBac transposase expression vector to shRNA or control plasmids, were incubated as described above and added to cortical precursors three hours after plating. For plasmid transfection of human

cortical precursors, Lipofectamine Stem Transfection Reagent (Invitrogen) was used as described by the manufacturer. Briefly, 1 µg of DNA and 1 µl of Lipofectamine Stem Transfection Reagent were prepared separately in 25 µl of Opti-MEM medium (Invitrogen), mixed, incubated for 30 min and added to precursors one day after plating. The target sequences for murine and human *Dhcr7* shRNA were 5'- GGAAGGTGCTTCTTGTTTA-3' and 5'- GGAAGTGGTTTGACTTCAA-3' respectively. The target sequence for the control shRNA was 5'-TCCCAACTGTCACGTTCTC-3'. For quantification, immunostaining and image acquisition were performed, and >100 cells per condition per experiment were counted and analyzed, and experiments were performed with 3 embryos per plasmid transfected and analyzed individually. For inhibition of MEK/ERK, PD98059 at 50 µM or Trametinib at 100 nM were added to precursors 24 hours after plating or transfection, and cultures were analyzed 3 days later. For depletion of cholesterol, β-cyclodextrin (at 0.5 mM, 2.5 mM and 5 mM) was added to the precursors for 30 min at 37 °C, washed with PBS, cultured with neural precursor medium and analyzed 3 days later. Quantification, immunostaining, and image acquisition were performed and >100 cells per condition per experiment were counted and analyzed, and experiments were performed with three embryos per genotype or per plasmid transfected individually.

## Neurosphere cultures

E13.5 cortices from *Dhcr7$^{+/+}$* or *Dhcr7$^{-/-}$* embryos were dissected and mechanically dissociated into a single-cell suspension by a fire-polished glass pipette as previously described (*Capecchi and Pozner, 2015*; *Tomita et al., 2020*). Cell density and viability were determined using trypan blue exclusion. Cells were seeded in triplicate at a clonal density of 10 cells/µl in 6 well (2 ml/well) ultra-low attachment culture plates (Coster) in serum-free medium supplemented with 20 ng/ml EFG (Sigma), 20 ng/ml FGF2 (Sigma), 2% B27 supplement (Invitrogen) and 2 µg/ml heparin (Sigma). Neurospheres were cultured for 6 days at 37 °C. To evaluate self-renewal potential, neurospheres were mechanically dissociated into single cell suspensions by fire-polished glass pipette, passed through a 45 µm nylon screen cell strainer, and cultured at a clonal density of 2 cells/µl for an additional 6 days.

## Immunocytochemistry and histological analysis

For morphometric analysis, immunostaining of tissue sections was performed as described (*Capecchi and Pozner, 2015*). Briefly, brain sections were permeabilized and blocked in PBST (1 X PBS, 0.5% (v/v) Triton X-100) containing 10% NGS for 1 hr. Brain slices were incubated with primary antibodies in PBST with 5% NGS at 4 °C overnight. The sections were incubated with secondary antibodies in PBST with 5% NGS for 1–2 hr at room temperature. Sections were counterstained with DAPI (Thermo Fisher Scientific). Slides were mounted in Fluoromount-G anti-fade reagent (Southern Biotech). Digital image acquisition was performed with EVO-FL Imagining System (Thermo Fisher Scientific). For quantification of precursor and neuron numbers, we analyzed sections at the medial-lateral level, counting all marker-positive cells in a 200 µm wide strip of the cortex, extending from the meninges to the ventricle. In all cases, we analyzed at least 3 similar cortical sections/embryo or pup from three different embryos or pups per genotype (for a total of at least 9 sections per genotype).

## Antibodies

The primary antibodies used for immunostaining were chicken anti-GFP (1:1000; Abcam), rabbit anti-Dhcr7 (1:100; Thermo Fisher Scientific), rabbit anti-Sox2 (1:200; Cell Signaling Technology), rabbit anti-Pax6 (1:1000; Covance), mouse anti-Ki67 (1:200; BD Biosciences), mouse anti-βIII-tubulin (1:1000; Covance), rabbit anti-βIII-tubulin (1:1000; Covance), rabbit anti-Tbr1 (1:500; Abcam), mouse anti-Satb2 (1:400; Abcam), rat anti-Ctip2 (1:500; Abcam), and rabbit anti-cleaved caspase 3 (1:400; Cell Signaling Technology), mouse anti-Nestin (1:1000; StemCell Technologies), rabbit anti-Brachyury (1:200; R&D Systems), goat anti-NCAM (1:200; R&D Systems), goat anti-Sox17 (1:200; R&D Systems), rabbit anti-Foxa2 (1:1000; Abcam). The secondary antibodies used for immunostaining were Rhodamine (TRITC)-conjugated goat anti-mouse and anti-rabbit IgG (1:500; Jackson ImmunoResearch Laboratories) and Alexa Fluor 488-conjugated goat anti-mouse, anti-rat and anti-rabbit IgG (1:800; Jackson ImmunoResearch Laboratories), Alexa Flour 488-conjugated donkey anti-goat IgG (1:800; Jackson ImmunoResearch Laboratories), Rhodamine (TRITC)-conjugated donkey anti-goat (1:200; Jaclson ImmunoResearch Laboratories). The primary antibodies used for immunoblotting were rabbit anti-Dhcr7 (1:1000; Abcam), rabbit anti-GAPDH (1:5000; Santa Cruz Biotechnology), rabbit anti Erk1/2

(1:5000; Santa Cruz Biotechnology), rabbit anti-TrkB (1:1000; Cell Signaling Technology), rabbit anti-phospho-TrkB (1:1000; Cell Signaling Technology), rabbit anti-glucocorticoid receptor (1:1000; Cell Signaling Technology), rabbit anti-phospho-glucocorticoid receptor (1:1000; Cell Signaling Technology), rabbit anti-MEK (1:1000; Cell Signaling Technology), rabbit anti-phospho-MEK (1:1000; Cell Signaling), rabbit anti-phospho-cebpβ (1:1000; Cell signaling Technology) and rabbit anti-cebpβ (1:1000; Cell Signaling Technology), rabbit anti-Pax6 (1:1000; Covance), mouse anti-Nestin (1:1000; StemCell Technologies), rabbit anti-Brachyury (1:200; R&D Systems), rabbit anti-Tbr2 (1:500; Abcam), goat anti-Sox17 (1:200; R&D Systems), rabbit anti-Foxa2 (1:1000; Abcam). The secondary antibodies used for immunoblotting were HRP-conjugated goat anti-mouse IgG (1:5000; Jackson ImmunoResearch Laboratories) and anti-rabbit IgG (1:10,000; Jackson ImmunoResearch Laboratories), HRP-conjugated donkey anti-goat IgG (1:1000; Jackson ImmunoResearch Laboratories).

## RT-PCR

Total RNA was isolated with Trizol and cDNA was prepared using the SuperScript III Reverse Transcriptase kit (Invitrogen) according to the manufacturer's protocols. Primer sequences are the following: *Dhcr7*-F: 5′-TATGAGGTGAATGGGCTGCA-3′; *Dhcr7*-R: 5′-GGTTAATGAGGGTCCAGGCT –3′; *β-actin*-F: 5′-GATGACGATATCGCTGCGCTG-3′; *β-actin*-R: 5′- GTACGACCAGAGGCATACAGG-3′. All PCR products were single bands with predicted molecular weights and confirmed by DNA sequencing.

## Quantitative PCR

Total RNA was extracted with Tri-Reagent (Sigma) treated with DNAse I (Fermentas, Thermo Scientific, Waltham, MA, USA) and cDNA was synthesized from 1 µg of RNA using the SuperScript IV Reverse Transcriptase Kit (Invitrogen) according to the manufacturer's protocols. Quantitative PCR was performed using Taqman Fast Advance Master Mix (Thermo Fisher Scientific) and Taqman probes targeted against either *Dhcr7* (Mm01164321_m1) or *β-Actin* (Mm00607939_s1). *β-Actin* mRNA was used as an endogenous control for all reactions, and all reactions were performed in triplicate. Quantitative PCR was performed and analyzed using StepOne Plus Real-Time PCR system (ThermoFisher Scientific).

## Western blotting

Embryonic cortices or neurosphere cultures were lysed in RIPA buffer (50 mM Tri pH8, 150 mM NaCl, 1% NP-40, 0.1% SDS, 1 mM EDTA) containing 1 mM PMSF (phenylmethanesulfonyl fluoride), 1 mM sodium vanadate, 20 mM sodium fluoride 10 µg/ml aprotinin and 10 µg/ml leupeptin. A total of 10–20 µg of protein lysate was electrophoresed, and western blots were performed as described previously (*Barnabé-Heider and Miller, 2003*).

## RNA sequencing and data analysis

Raw RNA sequencing reads in FASTQ format were mapped to the human genome using HISAT (https://ccb.jhu.edu/software/hisat/; Last accessed June 19, 2022), and format conversions were performed using Samtools. Cufflinks (http://cole-trapnell-lab.github.io/cufflinks/; Last accessed June 19, 2022) was used to estimate the relative abundances of transcripts from each RNA sample. Cuffdiff, a module of Cufflinks, was then used to determine differentially expressed genes (DEGs) between Control and *DHCR7*-KO hNPCs. DEGs met the following criteria: adjusted *P*-value < .05 (corresponding to the allowed false discovery rate of 5%) and fold-change >1.5 between genotypes. A two-way hierarchical clustering dendrogram (complete-linkage method, Euclidean distance scale) of DEGs was used to visualize biological variability among samples, generated by R software using the 'pheatmap' package (https://cran.r-project.org/web/packages/pheatmap/). To elucidate the biological functions of DEGs, the Core Analysis feature of Ingenuity Pathway Analysis (IPA, Qiagen) was used to identify significantly enriched Diseases and Biological Functions related to the nervous system. Network interactions among DEGs involved in the Biological Function 'development of the central nervous system' were assessed using STRING (Search Tool for the Retrieval of Interacting Genes/Proteins) analysis, set at the highest confidence interaction score and only connected nodes displayed (https://string-db.org/cgi/input.pl?sessionId=xCahIfrzvltC; Last accessed June 19, 2022). Enriched KEGG (Kyoto Encyclopedia of Genes and Genomes) pathways were identified among DEGs in the STRING network. Finally, a Venn diagram was generated to demonstrate the overlap between genes dysregulated in KO hNPCs

and genes in the SFARI database, a collection of genes implicated in autism susceptibility (https://gene.sfari.org/; Last accessed June 19, 2022). Raw data of the RNA sequencing has been deposited at Dryad (https://doi.org/10.5061/dryad.zw3r2287f). The list of differentially expressed genes can be found in *Supplementary file 2* (Excel).

## Oxysterol and sterol analysis

Cell pellets were resuspended in 300 µL of 1 X PBS and lysed by sonication in an ice bath for 30 min. Protein determination was performed using the BioRad DC protein mass assay (BioRad, Hercules, CA). Internal standard mixtures for sterols and oxysterols analysis were added to each sample (see *Supplementary files 1–4* for a list of standards and their concentrations used). Lipid extraction was performed using the Folch method as described previously (*Fliesler et al., 2018*; *Folch et al., 1957*; *Herron et al., 2018*). The dried lipid extract was reconstituted with 200 µL of methylene chloride and stored at –80 °C until analysis. Prior to analysis, 50 µL of extract was transferred to into glass LC vials, dried under argon, and reconstituted with 50 µL of 90% methanol in water with 0.1% formic acid. For tissue samples, sterol and oxysterol internal standard mixtures were added to whole tissues, which were subsequently homogenized in 4 mL Folch solution with 1 mL 0.9% NaCl. The dried lipid extract was reconstituted with 0.5 mL (for tissues < 50 mg), 1.0 mL (tissues > 50 < 100 mg) or 1.5 mL (>100 mg) of methylene chloride. Prior to analysis, 50 µL of lipid extract (30 µL for tissues > 100 mg) was transferred into glass LC vials, dried under argon, and reconstituted with 50 µL 90% methanol in water with 0.1% formic acid. Determination of oxysterol and sterol concentrations in cells and tissues was performed by ultra-performance liquid chromatography (UPLC) tandem mass spectrometry (MS/MS) on a SCIEX 6500 triple quadrupole mass spectrometer (for oxysterols) or a SCIEX 4000 QTRAP (for sterols) mass spectrometer with atmospheric pressure chemical ionization (APCI) coupled to a Waters Acquity UPLC system, as described previously (*Fliesler et al., 2018*; *Herron et al., 2018*). Briefly, sterols and oxysterols were separated by reverse-phase chromatography on a C18 column (1.7 mm, 2.1x100 mm, Phenomenex Kinetex) using a 15 min isocratic gradient of 90% methanol with 0.1% formic acid at a flow of 0.4 mL/min. Selective reaction monitoring (SRM) was used to monitor the dehydration of the sterol and oxysterol $[M+H]^+$ ions to generate $[M+H-H_2O]^+$ ions (See *Supplementary files 1-4* for a list of retention times and MS/MS transitions for the standards used). The APCI parameters were as follows: nebulizer current, 3 mA; temperature, 350 °C; curtain gas, 20 psi; ion source gas, 55 psi. The MS conditions for SRM analysis were as follows: declustering potential, 80 V; entrance potential, 10 V; collision energy, 25 V; collision cell exit potential, 20 V. Data analysis was performed with Analyst (v. 1.6.2) Quantitation Wizard. Analyte concentrations in cells and tissues were determined relative to the internal standard levels and the relative response factor (RRF) of each analyte was calculated from a mixture of sterol and oxysterol standards and internal standards. Concentrations were normalized to cell protein weight or tissue weight.

## Docking simulation

Docking simulations were performed with Autodock Vina 1.1.2. The model ligands and 3D models were built and generated by Openbabel 2.3.2. The crystal structure of the glucococorticoid receptor used for docking simulations was crystal structure 4p6w available from Protein Data Bank (http://www.rcsb.org/structure/4p6w). The mometasone furoate molecule present in that crystal structure was used to establish the center of the search area defining the binding pocket, and ligand docking calculations were performed within that search area.

## Statistical analysis

In general, three biological replicates were used for each experiment based on the strong phenotype observed in the *Dhcr7⁻/⁻* SLOS model and our previous publication showing the large differences in oxysterols and sterol levels in WT and KO brains at birth (*Xu et al., 2011*). For culturing of mouse cortical precursors or neurospheres from single embryos, one litter is considered a biological replicate. For hNPC culture, each separate preparation of hNPC from hiPSC is considered a biological replicate. For in vivo study, one litter is considered a biological replicate. Statistical analyses were performed using two-tailed Student's *t*-test assuming unequal variance when comparing two different groups unless otherwise indicated in the text. For immunostaining of cell culture or tissue sections, at least three technical replicates per biological replicate were performed. To analyze the multi-group

neuroanatomical studies, we used one-way ANOVA unless otherwise indicated in the text. Significant interactions or main effects were further analyzed using Newman-Keuls post-hoc tests. All statistical tests were performed using Microsoft Excel or Prism 8 (GraphPad). In all cases, error bars indicate the standard error of the mean.

## Acknowledgements

This work was supported by a National Institutes of Health (NIH) grant R01HD092659 (LX) and a grant from the Smith-Lemli-Opitz/RSH Foundation. JMH was supported by the University of Washington (UW) Environmental Pathology/ Toxicology Training Program (NIH T32 ES007032) and AL was supported by the UW Pharmacological Sciences Training Program (NIH T32 GM007750) and the Institute of Translational Health Sciences TL1 Program (NIH TL1 TR002318). We also thank Dr. Carol Ware and Christopher Cavanaugh for thier technical advice on hiPSC culture.

## Additional information

### Funding

| Funder | Grant reference number | Author |
|---|---|---|
| National Institutes of Health | R01 HD092659 | Libin Xu |
| National Institutes of Health | T32 ES007032 | Josi M Herron |
| National Institutes of Health | T32 GM007750 | Amy Li |
| National Institutes of Health | TL1 TR002318 | Amy Li |
| Smith-Lemli-Opitz/RSH Foundation | Research grant | Libin Xu |

The funders had no role in study design, data collection and interpretation, or the decision to submit the work for publication.

### Author contributions

Hideaki Tomita, Conceptualization, Data curation, Formal analysis, Validation, Investigation, Visualization, Methodology, Writing - original draft, Project administration; Kelly M Hines, Josi M Herron, David W Baggett, Data curation, Formal analysis, Validation, Investigation, Visualization, Methodology, Writing – review and editing; Amy Li, Data curation, Formal analysis, Validation, Investigation, Writing – review and editing; Libin Xu, Conceptualization, Resources, Formal analysis, Supervision, Funding acquisition, Visualization, Project administration, Writing – review and editing

### Author ORCIDs

Amy Li ⓘ http://orcid.org/0000-0002-7732-3540
Libin Xu ⓘ http://orcid.org/0000-0003-1021-5200

### Ethics

This study was performed in strict accordance with the recommendations in the Guide for the Care and Use of Laboratory Animals of the National Institutes of Health. All of the animals were handled according to approved institutional animal care and use committee (IACUC) protocols (#4350-01) of the University of Washington.

### Decision letter and Author response

Decision letter https://doi.org/10.7554/eLife.67141.sa1
Author response https://doi.org/10.7554/eLife.67141.sa2

# Additional files

## Supplementary files

• Supplementary file 1. Retention times and MS/MS transitions for oxysterol internal standards. Relate to *Figure 2* and Figure 2-Figure Supplement 2.

• Supplementary file 2. Retention times and MS/MS transitions for all oxysterol standards. Relate to *Figure 2* and *Figure 2—figure supplement 2*.

• Supplementary file 3. Retention times and MS/MS transitions for sterol internal standards. Relate to *Figure 2* and *Figure 2—figure supplement 2*.

• Supplementary file 4. Retention times and MS/MS transitions for sterol standards. Relate to *Figure 2* and *Figure 2—figure supplement 2*.

• Supplementary file 5. Ingenuity Pathway Analysis (IPA) reveals "development of the central nervous system" as one of the top 10 enriched Diseases and Biological Functions related to the nervous system. Relate to *Figure 7*. DEGs were further analyzed with (IPA) to identify the most enriched biological functions related to the nervous system in SLOS mutant NPCs. The table below shows the top ten enriched terms corresponding to Diseases/Bio-functions along with the *p*-value and overlapping number of genes in the dataset.

• Supplementary file 6. Excel file. List of differentially expressed genes in SLOS hNPCs relative to Control hNPCs. Relate to *Figure 7*.

• Transparent reporting form

## Data availability

Raw RNA sequencing data has been deposited at Dryad at https://doi.org/10.5061/dryad.zw3r2287f. This data was used to generate Figure 7 and Table S5.

The following dataset was generated:

| Author(s) | Year | Dataset title | Dataset URL | Database and Identifier |
|---|---|---|---|---|
| Xu L, Tomita H, Herron J | 2021 | 7-Dehydrocholesterol-derived oxysterols cause neurogenic defects in Smith-Lemli-Opitz syndrome | https://dx.doi.org/10.5061/dryad.zw3r2287f | Dryad Digital Repository, 10.5061/dryad.zw3r2287f |

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
