## [Editor Report]

This is an important paper that provides a conceptual framework for understanding how altered lipid metabolism can impact brain development. The authors use mouse models and human iPSCs to provide a convincing mechanistic explanation of how mutations in a key enzyme in cholesterol biosynthesis lead to a neurodevelopmental disorder.

---

## [Decision Letter]

**Decision letter after peer review:**

Thank you for submitting your article "7-Dehydrocholesterol-derived oxysterols cause neurogenic defects in Smith-Lemli-Opitz syndrome" for consideration by *eLife*. Your article has been reviewed by 3 peer reviewers, including Anita Bhattacharyya as the Reviewing Editor and Reviewer #1, and the evaluation has been overseen by Huda Zoghbi as the Senior Editor.

Reviewers agree that the study is of high interest in that it provides a causal link between the DHCR7 depletion in SLOS and neurogenesis defects and thus could have a critical impact on SLOS therapy. The foundational data that show that LOF of DHCR7 leads to premature differentiation of cortical progenitors is convincing and the ability to phenocopy with exogeneous oxysterols lends strength to the study and links to the human disorder SLOS. The use of mouse models and validation in human cells is a strength.

However, the manuscript in its current form does not meet the standards for publication in *eLife*. Reviewers identified specific areas that require improvement to support the claims of the paper and to offer new mechanistic insight.

1. The authors have not convincingly distinguished proliferation from enhanced differentiation. Showing the number of Ki67 positive cells is a relatively weak way of demonstrating reduced neurogenesis. BrdU labelling or similar approach with quantification should be used. In addition, the description of the effects altered neurogenesis is somewhat imprecise. "Altered cytoarchitecture" and "proper localization" suggest that an alternative hypothesis may be that cells do not migrate properly, which is not tested in this manuscript. In cortical neurogenesis, it seems logical that if progenitors prematurely differentiate, then they will not populate the more superficial layers. This is not explicitly stated.

2. Protein analysis is weak. The western blots have no molecular weight markers, no control proteins, and no quantification that is needed to support claims of over or under expression (Figures 1C, 3A and 3M, 5J, 7A and 7C). Quantification of Western blots is needed to support the claims of altered expression throughout the manuscript.

3. The RNAseq experiments give interesting gene expression results and reveal possible signaling pathways involved. However, further data is needed to support the central role of TrkB, as the rationale for the focus on TrkB is weak (many growth factors signal through MapK) and the data in Figure 5J are not convincing. Many signaling pathways could account for the phenotype and those regulated by the cholesterol pathway would be stronger candidates. Further, the RNA seq and results in Figure 7 are difficult to follow as they alternate between human and mouse. Do the RNA seq results in human agree with mouse models?

4. The use of both mouse and human models is a major strength of the work to confirm that phenotypes in mouse models are relevant to patients with SLOS. However, the human data is relegated to supplementary data, thus limiting its impact. The data should be included in the main manuscript.

---

## [Author Response]

The reviewers have discussed their reviews with one another, and the Reviewing Editor has drafted this to help you prepare a revised submission.Reviewers agree that the study is of high interest in that it provides a causal link between the DHCR7 depletion in SLOS and neurogenesis defects and thus could have a critical impact on SLOS therapy. The foundational data that show that LOF of DHCR7 leads to premature differentiation of cortical progenitors is convincing and the ability to phenocopy with exogeneous oxysterols lends strength to the study and links to the human disorder SLOS. The use of mouse models and validation in human cells is a strength.However, the manuscript in its current form does not meet the standards for publication in eLife. Reviewers identified specific areas that require improvement to support the claims of the paper and to offer new mechanistic insight.1. The authors have not convincingly distinguished proliferation from enhanced differentiation. Showing the number of Ki67 positive cells is a relatively weak way of demonstrating reduced neurogenesis. BrdU labelling or similar approach with quantification should be used. In addition, the description of the effects altered neurogenesis is somewhat imprecise. "Altered cytoarchitecture" and "proper localization" suggest that an alternative hypothesis may be that cells do not migrate properly, which is not tested in this manuscript. In cortical neurogenesis, it seems logical that if progenitors prematurely differentiate, then they will not populate the more superficial layers. This is not explicitly stated.

We thank the reviewer for the suggestion. We have now carried out EdU labeling experiments in cortical precursors (Figure 3F), which showed that *Dhcr7*-knock down leads to reduced proliferation of the cortical precursors. We also want to point out that in vivo EdU labelling experiments with quantification were performed in our initial submission, which showed increased cell cycle exit and decreased proliferation index of cortical precursors in *DHCR7* mutants (Figure 6).

We have also modified the wording “altered cytoarchitecture” to “reduced thickness of cortical layers” to more accurately reflect our observations in Figure 5. We tracked the formation of neurons from E12.5 to E15.5 using EdU, which confirmed the premature neurogenesis phenotype in vivo (Figure 5H-I). However, E15.5 is not the end of the cortical neurogenesis, so we do not expect to see reduced neurons in the superficial layers at this time point yet. Our data in Figure 3 and 6A-D suggests that *Dhcr7* knockdown or KO leads to depletion of the cortical precursor pools, which ultimately leads to reduced thickness of VZ/SVZ at E14.5 and E15.5 (Figure 6E-F) and reduced thickness of cortical layers at E18.5 (Figure 5A-C). On the other hand, no obvious migration defect in *Dhcr7*^-/-^ neurons was observed in Figure 5H and 6A as EdU-labelled cells appear in layers similar to those in wild-type.

2. Protein analysis is weak. The western blots have no molecular weight markers, no control proteins, and no quantification that is needed to support claims of over or under expression (Figures 1C, 3A and 3M, 5J, 7A and 7C). Quantification of Western blots is needed to support the claims of altered expression throughout the manuscript.

We thank the reviewer for pointing this out. We have included control proteins in all the western blot analysis: Erk1/2 for Figures 1C, Erk1/2 for Figures 3A, 3N, and 8C (note the numbering change; previous Figure 7 is now Figure 8), GAPDH for Figure 5J, and 8A, 8M, and 8O. The western blots in Figures 1C, 3A, 3N, 8C, 8M, and 8O showed the targeting proteins either presence or nearly invisible, so we do not feel the needs for absolute quantitation. However, to demonstrate the changes in phosphorylation levels of the neurogenesis signaling pathway, we have quantified the phosphorylated forms of various proteins in Figure 5J and 8A and added the quantitation plots (Figure 5K and 8A).

3. The RNAseq experiments give interesting gene expression results and reveal possible signaling pathways involved. However, further data is needed to support the central role of TrkB, as the rationale for the focus on TrkB is weak (many growth factors signal through MapK) and the data in Figure 5J are not convincing. Many signaling pathways could account for the phenotype and those regulated by the cholesterol pathway would be stronger candidates. Further, the RNA seq and results in Figure 7 are difficult to follow as they alternate between human and mouse. Do the RNA seq results in human agree with mouse models?

We thank the reviewer for the critical suggestion. Our main finding for the disease mechanism is that one of the 7-DHCderived oxysterols, DHCEO can activate glucocorticoid receptor (GR) and GR can hyperactivate TrkB, which is the main regulator of embryonic neurogenesis. Our RNAseq experiments were intended to show the activation of GR-TrkB further hyper-activates downstream effectors of the neurogenic signaling pathway such as MapK, which was indeed found to be a major affected pathway (new Figure 7; Table S5). The RNAseq experiments did not find any neurogenic candidate that are regulated by the cholesterol pathway.

To strength our rationale for the involvement of GR-TrkB activation to the premature neurogenesis phenotypes of SLOS, we have performed a new series of *DHCR7* knockdown experiments with TrkB or MEK knockdown or MEK inhibitors (new Figures 8K-O), which showed that the premature neurogenesis was reversed when either TrkB or its downstream effector MEK were blocked. This supports the involvement of this particular neurogenic pathway in the observed premature neurogenesis phenotype.

4. The use of both mouse and human models is a major strength of the work to confirm that phenotypes in mouse models are relevant to patients with SLOS. However, the human data is relegated to supplementary data, thus limiting its impact. The data should be included in the main manuscript.

We have re-organized our manuscript so that the date on human SLOS model is a part of the main manuscript (see new Figure 1 – Panels P-S).